# Exploring Federated Pruning for Large Language Models

## Abstract

LLM pruning has emerged as a promising technology for compressing LLMs, enabling their deployment on resource-limited devices. However, current methodologies typically require access to public calibration samples, which can be challenging to obtain in privacy-sensitive domains. To address this issue, we introduce **FedPrLLM**, a comprehensive federated pruning framework designed for the privacy-preserving compression of LLMs. In FedPrLLM, each client only needs to calculate a pruning mask matrix based on its local calibration data and share it with the server to prune the global model. This approach allows for collaborative pruning of the global model with the knowledge of each client while maintaining local data privacy. Additionally, we conduct extensive experiments to explore various possibilities within the FedPrLLM framework, including different comparison groups, pruning strategies, and the decision to scale weights. Our extensive evaluation reveals that one-shot pruning with layer comparison and no weight scaling is the optimal choice within the FedPrLLM framework. We hope our work will help guide future efforts in pruning LLMs in privacy-sensitive fields. Our code is available at https://anonymous.4open.science/r/FedPrLLM-15594.

## 1 Introduction

Large Language Models (LLMs) (Brown, 2020; Touvron et al., 2023a; Achiam et al., 2023) have revolutionized the field of natural language processing by demonstrating remarkable capabilities across various tasks. However, their increasing size leads to significant hardware requirements, limiting real-world deployment. To address this, research has focused on compact LLMs through compression techniques, such as *pruning* (Ma et al., 2023; Frantar & Alistarh, 2023; Sun et al., 2024), *knowledge distillation* (Gu et al., 2024; Xu et al., 2024b), *quantization* (Xiao et al., 2023; Shao et al., 2023), and *low-rank factorization* (Zhao et al., 2024; Saha et al., 2023). Among these, pruning has emerged as a promising method to reduce resource demands by selectively removing redundant parameters while preserving performance (Ma et al., 2023; Frantar & Alistarh, 2023). Typically, LLM pruning methods can be broadly classified into *structured pruning*, which removes entire substructures within LLMs, such as neurons (Ma et al., 2023; Li et al., 2023; Ashkboos et al., 2024), layers (Xia et al., 2024), or even entire transformer blocks (Gromov et al., 2025), and *unstructured pruning*, which removes individual weights from the model's weight matrices based on certain criteria (Frantar & Alistarh, 2023; Sun et al., 2024; Zhang et al., 2024b; Yin et al., 2024; Xu et al., 2024a). This work focuses on unstructured pruning, as it tends to achieve higher compression rates and maintain better model performance compared to structured pruning (Frantar & Alistarh, 2023; He et al., 2024; Xia et al., 2024; Zhang et al., 2024b).

Despite advances in LLM unstructured pruning methods, these approaches usually rely on access to public calibration data to guide the pruning process (Frantar & Alistarh, 2023; Sun et al., 2024; Zhang et al., 2024b; Yin et al., 2024; Xu et al., 2024a). Specifically, they require calibration samples to evaluate the importance of the model weights in order to determine the pruning mask matrix for pruning models. However, in many real-world scenarios, such as healthcare, finance, and personalized services, the data used for pruning might be private and cannot be shared due to privacy regulations and concerns. Federated Learning (FL) (McMahan et al., 2017; Zhang et al., 2024a; Zeng et al., 2024; Guo et al., 2025b;a), which utilizes collaborative and decentralized training of models across multiple institutions without sharing personal data externally, offers a promising solution to this challenge.

Integrating FL with LLM pruning allows each client to calculate a local pruning mask matrix based on its private calibration data and share it with the server. The server then aggregates these mask matrices into an aggregated mask matrix and selects the top-k values (the most clients want to prune) to derive a final pruning mask matrix for pruning the global model. Despite its ability to protect data privacy, three unresolved challenges within this framework hinder practical deployment.

**Challenge 1: How to compare parameters?** When selecting the top-k values, a critical ambiguity arises: Should parameter importance be compared across the entire layer or within each respective row or column (corresponding to *layer*, *row*, and *column comparisons*, respectively)? Previous centralized LLM pruning work (Sun et al., 2024) has highlighted the importance of using a proper comparison group for pruning LLMs, yet no study explores this in federated scenarios.

**Challenge 2: To scale or not scale for retained parameters.** Beyond simply determining which parameters to prune via majority voting (i.e., selecting top-k values), the FL aggregated mask matrix reveals a critical hidden signal: how strongly each parameter is disfavored across clients. Consider two surviving parameters - one narrowly retained (pruned by 10/100 clients) and another unanimously preserved (pruned by 0/100 clients). Traditional pruning treats both equally, maintaining their original magnitudes despite their differing consensus levels. However, this ignores a critical insight: the former parameter, though retained, exhibits weaker consensus across clients. This observation raises a fundamental question: Rather than simply employing binary masking, could we leverage the FL aggregated mask matrix to guide continuous weight adjustment, where retained parameters are scaled down proportionally based on their pruning frequency?

**Challenge 3: Is iterative pruning worth the cost?** LLM pruning is typically performed *layer-by-layer* recursively to avoid error accumulation (Frantar & Alistarh, 2023; Sun et al., 2024; Zhang et al., 2024b). As a result, in FL, this necessitates either *one-shot pruning* (clients compute all layer mask matrices and share them with the server in one go) or *iterative pruning* (clients send the mask matrices to the server layer by layer in an iterative manner). While iterative pruning allows for refining the local model promptly, it incurs prohibitive communication costs for deep LLMs. This raises an unstudied question: Does iteratively refining the local model improve accuracy enough to justify its massive communication overhead?

To address these challenges, we formalize the first systematic and comprehensive empirical study of the fundamental design space of federated LLM pruning and empirically evaluate three core design choices through a unified **FedPrLLM** framework (Figure 1):

**Q1.** *Comparison Group*: Which comparison group is more effective: *layer*, *row*, or *column*?

**Q2.** *Weight Scaling*: Should we scale the model weights of the retained parameters?

**Q3.** *Pruning Strategy*: Does iterative pruning outperform one-shot pruning?

We dedicated thousands of GPU hours to benchmark federated pruning for LLMs, conducting extensive experiments across **6** open-source LLMs, **4** local pruning methods, **3** sparsity ratios, **3** comparison groups, **2** pruning strategies on **10** common datasets. From these efforts, we have developed a practical list of key insights for federated pruning of LLMs:

1). **Layer comparison is simple yet effective.** Among the three comparison groups—*layer*, *row*, and *column comparisons*—layer comparison stands out as the simplest and most effective method, regardless of the local pruning method's comparison group.

2). **Scaling weights performs worse than expected.** Though the FL aggregated mask matrix, which reveals how strongly each parameter is disfavored across clients, could be used to scale the retained parameters for continuous weight adjustment, its performance is inferior to that of not scaling them.

3). **Iterative pruning offers no benefit.** While iterative pruning allows for prompt refinement of the local model, it incurs significant communication overhead, and its performance is comparable to that of one-shot pruning, offering no additional advantages.

We hope our findings will help guide future efforts in federated pruning for LLMs and inform best practices for deploying LLMs under federated scenarios in real-world applications. We summarize our contributions as follows:

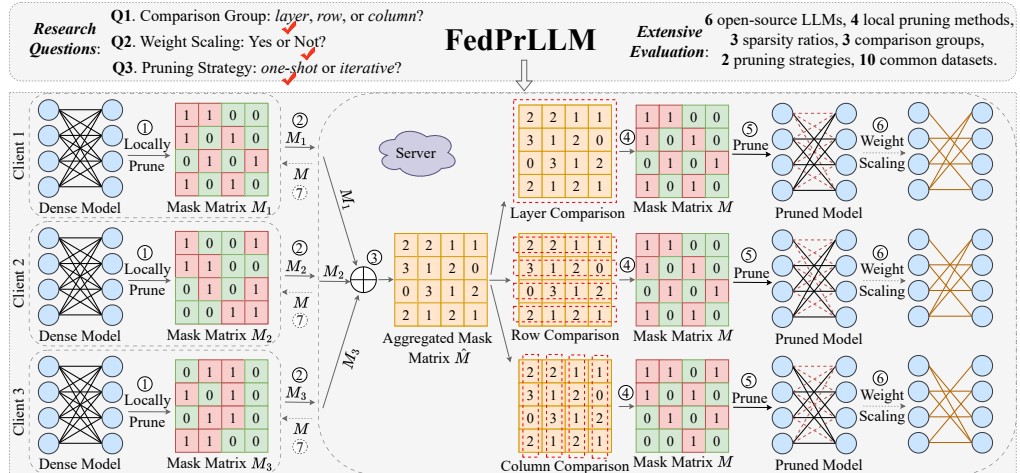

Figure 1: *Top)*. Research questions alongside the corresponding findings and experimental scenarios. *Bottom)*. The FedPrLLM framework. ① Each client calculates a pruning mask matrix $\mathbf{M}_i$ using its calibration dataset $\mathcal{D}_i$. ② Clients send the mask matrices $\mathbf{M}_i$ to the server. ③ The server aggregates these mask matrices $\mathbf{M}_i$ to obtain an aggregated mask matrix $\hat{\mathbf{M}} = \sum_{i=1}^{m} \mathbf{M}_i$. ④ Top-k values are selected from the aggregated mask matrix $\hat{\mathbf{W}}$ to derive the final mask matrix $\mathbf{M}$. ⑤ Prune the global model $\mathbf{W}$ using the mask matrix $\mathbf{M}$ as follows: $\hat{\mathbf{W}} = \mathbf{W} \odot (1 - \mathbf{M})$, where $\odot$ denotes element-wise multiplication. ⑥ Scale the model weights of the retained parameters using the aggregated mask matrix $\hat{\mathbf{M}}$ as follows: $\hat{\mathbf{W}} \odot \frac{(m - \hat{\mathbf{M}})}{m}$ (if needed). ⑦ The server broadcasts the mask matrix $\mathbf{M}$ to each client (for iterative pruning). The dashed arrow indicates that this operation is optional; step ⑥ is used for weight scaling, while ⑦ is used for iterative pruning. Note that this visualization is primarily for one-shot pruning, which requires only one communication round. For iterative pruning, multiple communication rounds will occur between steps ② and ⑦, and the layer index is omitted here.

- We introduce **FedPrLLM**, a comprehensive federated pruning framework designed for the privacy-preserving compression of LLMs, which incorporates various possibilities for integrating FL with LLM pruning.

- We conduct an extensive evaluation of FedPrLLM, providing practical insights into effective federated pruning techniques for LLMs, based on thousands of GPU hours invested in multiple open-source LLMs, various sparsity ratios, comparison groups, and datasets.

- We identify that layer comparison is simple yet effective, scaling weights offers no benefits and may worsen performance, and that one-shot pruning is as effective as iterative pruning while reducing communication costs.

## 2 PRELIMINARIES

In this section, we review some concepts related to LLM pruning. LLM pruning can be broadly classified into *structured pruning* (Ma et al., 2023; Li et al., 2023; Ashkboos et al., 2024; Xia et al., 2024; Gromov et al., 2025) and *unstructured pruning* (Frantar & Alistarh, 2023; Sun et al., 2024; Zhang et al., 2024b; Yin et al., 2024; Xu et al., 2024a), and in this work, we focus on the latter. Unstructured pruning involves removing individual weights from the model's weight matrices based on certain criteria while maintaining its performance as much as possible (Frantar & Alistarh, 2023; Sun et al., 2024; Zhang et al., 2024b; Yin et al., 2024; Xu et al., 2024a). It is usually achieved by minimizing the discrepancy square error between the dense and pruned model *layer-by-layer* recursively. Specifically, for an uncompressed linear layer with weights $\mathbf{W}_l \in \mathbb{R}^{d \times r}$, the objective for unstructured pruning can usually be formulated as:

$$\arg\min_{\mathbf{M}_l} \|\mathbf{W}_l \mathbf{X}_l - (\mathbf{W}_l \odot (1 - \mathbf{M}_l))\mathbf{X}_l\|_2^2 \quad \text{s.t.} \quad \|\mathbf{M}_l\|_0 = k, \tag{1}$$

where $\mathbf{X}_l$ is the input to $l$-th linear layer (also referred to as calibration data), $\mathbf{M}_l \in \{0,1\}^{d \times r}$ is the pruning mask matrix we aim to derive, $\odot$ denotes element-wise multiplication, $\|\cdot\|_0$ is the $l_0$-norm (e.g., the number of non-zero elements), and $k$ represents the number of pruned weights determined by the pruning ratio.

The differences between previous pruning methods primarily lie in the design of the pruning metrics and the comparison groups used to derive the pruning mask matrix (Frantar & Alistarh, 2023; Sun et al., 2024; Zhang et al., 2024b). Pruning metrics refer to how the importance of each model weight is identified, while comparison groups denote the selection of groups for comparing these weights, including *layer comparison*, *row comparison*, and *column comparison*. For example, SparseGPT (Frantar & Alistarh, 2023) utilizes the Hessian Matrix inverse, i.e., $\left[ \frac{|\mathbf{W}|^2}{\text{diag}\left((\mathbf{X}^\mathsf{T}\mathbf{X}+\lambda\mathbf{I})^{-1}\right)} \right]_{ij}$, as the pruning metric, employing layer comparison to determine the pruning mask matrix for pruning, along with subsequent weight scaling. Wanda (Sun et al., 2024) adopts the magnitudes of model weights multiplied by the corresponding input activations, i.e., $|\mathbf{W}_{ij}| \cdot \|\mathbf{X}_j\|_2$, as the pruning metric and chooses row comparison.

## 3 FEDERATED PRUNING FOR LLMS

### 3.1 PROBLEM FORMULATION

In the federated pruning scenario for LLMs, multiple clients aim to collaboratively prune an LLM while ensuring that their local calibration data remains private. Formally, let $\mathbf{W}$ represent the model parameters of the LLM that we aim to prune. Each client $i$ possesses a private calibration dataset denoted as $\mathcal{D}_i$, which is used for calculating the pruning mask matrices during the local pruning process. These mask matrices are then shared with the server to prune the LLM.

### 3.2 FEDPRLLM

In this section, we first introduce the overall workflow of the comprehensive **FedPrLLM** framework, as illustrated at the bottom of Figure 1, and then discuss the various possibilities within it. Specifically, during local pruning, each client calculates a pruning mask matrix $\mathbf{M}_i \in \{0,1\}^{|\mathbf{W}_i|}$ using its calibration dataset $\mathcal{D}_i$ (step ①). This mask matrix determines which weights are pruned ($\mathbf{M}_{ij} = 1$) and which are retained ($\mathbf{M}_{ij} = 0$). The decision on which weights to prune or retain is based on an importance criterion derived from the calibration data, such as the magnitudes of model weights multiplied by the corresponding input activations used in Wanda (Sun et al., 2024) or other pruning methods.

After calculating the pruning mask matrix, each client $i$ shares only the mask matrix $\mathbf{M}_i$ with the central server (step ②). This approach ensures that no local model parameters or private calibration data are transmitted, thereby minimizing communication overhead and preserving data privacy. Upon receiving the pruning mask matrices $\mathbf{M}_i$ from all clients, the server sums them to obtain an aggregated mask matrix $\hat{\mathbf{M}} = \sum_{i=1}^{m} \mathbf{M}_i$ (step ③) and then selects the top-k values to create the final mask matrix $\mathbf{M}$ (step ④)[1] for pruning the global model (step ⑤). In the following, we will discuss various possibilities within the FedPrLLM framework, including different comparison groups, the decision to perform weight scaling, and the choice between one-shot and iterative pruning.

#### 3.2.1 COMPARISON GROUP

When selecting the top-k values from the aggregated mask matrix $\hat{\mathbf{M}}$ to derive the final pruning mask matrix $\mathbf{M}$, three comparison groups can be considered (step ④): *layer comparison*, *row comparison*, and *column comparison*. In layer comparison, the comparison group consists of all elements within a layer, allowing us to choose the top-k values across the entire layer. Conversely, in row (or column) comparison, the comparison group is defined by each individual row (or column), enabling the selection of the top-k values within each respective row (or column). The visualization of these comparison groups is shown in Figure 1. Thus, given that multiple comparison groups could be chosen, *which comparison group is more effective for federated pruning of LLMs?*

---

[1]The rationale behind such voting mechanism is shown in Section A in Appendix.

### 3.2.2 Weight Scaling

After obtaining the final mask matrix $\mathbf{M}$, it can be used to effectively prune the dense model $\mathbf{W}$ using $\mathbf{W} \odot (1 - \mathbf{M})$, where $\odot$ denotes element-wise multiplication (step ⑤). This operation removes the weights corresponding to the masked parameters (i.e., $\mathbf{M}_{ij} = 1$), resulting in a sparser model $\hat{\mathbf{W}}$.

Then, beyond merely determining which parameters to prune via majority voting (i.e., selecting top-k values), the aggregated mask matrix $\hat{\mathbf{M}}$ reveals a critical hidden signal: how strongly each parameter is disfavored across clients. Consider two surviving parameters - one narrowly retained (pruned by 10/100 clients) and another unanimously preserved (pruned by 0/100 clients). Traditional pruning treats both equally, maintaining their original magnitudes despite their differing consensus levels. However, this ignores a critical insight: the former parameter, though retained, exhibits weaker consensus across clients. To this end, the aggregated mask matrix $\hat{\mathbf{M}}$ could be further applied to scale down the retained parameters using the formula $\hat{\mathbf{W}} \odot \frac{(m - \hat{\mathbf{M}})}{m}$ (step ⑥, if needed). This approach corresponds to locally pruning the model and then sharing the pruned model with the server, which aggregates them using the FedAvg algorithm (McMahan et al., 2017). However, *will the weight scaling improve the performance of federated pruning for LLMs*?

### 3.2.3 One-shot vs. Iterative Pruning

Since LLMs are usually pruned *layer-by-layer* recursively (Frantar & Alistarh, 2023; Sun et al., 2024; Zhang et al., 2024b), federated pruning for LLMs can be naturally categorized into two types: *one-shot pruning* and *iterative pruning*. In one-shot pruning, each client calculates the pruning mask matrices for all layers and then sends them to the server, resulting in only one communication round. In contrast, iterative pruning involves sending the pruning mask matrices to the server layer by layer. Specifically, after calculating the pruning mask matrix for one layer, it is uploaded to the server for aggregation. The server then combines these matrices into a global mask matrix for pruning the model at that layer and broadcasts the global mask matrix back to each client for local pruning of that layer (step ⑦, the layer index is omitted here). This process is carried out layer by layer and involves multiple communication rounds, resulting in higher communication costs compared to one-shot pruning. Therefore, given the significant communication costs associated with iterative pruning, *will iterative pruning outperform one-shot pruning*?

One-shot and iterative pruning differ because, when calculating the pruning mask matrix for layer $l + 1$ locally, the calibration data $\mathbf{X}_{l+1}$ is derived from the output of layer $l$, which has already been pruned. Since the weights of the local pruned model for layer $l$ vary between using $\mathbf{M}_i$ (one-shot pruning) and $\mathbf{M}$ (iterative pruning), this leads to different outputs for layer $l$ and, consequently, varying calibration data $\mathbf{X}_{l+1}$, resulting in distinct pruning mask matrices for layer $l + 1$.

## 4 Experiments

Our experiments are designed to answer the following research questions that are important for the practical pruning of LLMs under a federated scenario.

- **Q1.** Which comparison group is more effective: *layer*, *row*, or *column*?
- **Q2.** Should we scale the model weights of the retained parameters?
- **Q3.** Does iterative pruning outperform one-shot pruning?

### 4.1 Experimental Setup

We implement FedPrLLM in PyTorch (Paszke et al., 2019) and use the Hugging Face Transformers library (Wolf et al., 2019) to handle models and datasets. We evaluate the FedPrLLM on the three most widely adopted LLM model families: LLaMA 7B/13B/30B (Touvron et al., 2023a), LLaMA-2 7B/13B (Touvron et al., 2023b) and LLaMA-3 8B (Meta, 2024). For each model under consideration, we focus on pruning the linear layers (skipping the first embedding layer and the final classification head), which account for around 99% of the total LLM parameters. We employ unstructured sparsity and impose a uniform sparsity ratio for all linear layers.

For the calibration data, following (Frantar & Alistarh, 2023; Sun et al., 2024; Xu et al., 2024a; Zhang et al., 2024b), we use 128 samples from the C4 dataset (Raffel et al., 2020), with each sample containing 2048 tokens. For FedPrLLM, we set the number of clients to 64, resulting in each client having only 2 calibration samples. For each client, we adopt Wanda (Sun et al., 2024) SparseGPT (Frantar & Alistarh, 2023), OWL (Yin et al., 2024), and BESA (Xu et al., 2024a) to perform local pruning and calculate the pruning mask matrix.

Apart from the proposed FedPrLLM framework, we further implement two baselines for comparison: (1) **Local-only**, where each client prunes the model locally using its private calibration data, and (2) **Centralized**, where the server prunes the model with all calibration data, which could be considered as an upper bound for the pruning performance under FL setting.

Following previous works on LLM compression (Frantar & Alistarh, 2023; Xu et al., 2024a; Zhang et al., 2024b), we measure the performance of pruned models in language modeling and evaluate their perplexity on the held-out WikiText2 (Merity et al., 2017) validation set, C4 (Raffel et al., 2020) validation data, and PTB (Marcus et al., 1994). For further evaluation, we also assess the pruned models on seven zero-shot tasks from lm-evaluation-harness[2]: BoolQ (Clark et al., 2019), RTE (Wang et al., 2018), HellaSwag (Zellers et al., 2019), WinoGrande (Sakaguchi et al., 2021), ARC Easy and Challenge (Clark et al., 2018), and OpenbookQA (Mihaylov et al., 2018). The evaluation metric is accuracy.

## 4.2 MAIN RESULTS

To answer the research questions above, we conducted extensive experiments to evaluate FedPrLLM along with two baselines across **6** open-source LLMs, **4** local pruning methods, **3** sparsity ratios, **3** comparison groups, **2** pruning strategies on **10** common datasets. The experimental results using Wanda as the local pruning method for the 50% sparsity ratio on the WikiText2 dataset are shown in Table 1, while results for higher sparsity ratios (e.g., 60% and 70%) and other datasets (e.g., C4 and PTB) are shown in Tables 6, 7, and 8 in Appendix. More results using SparseGPT, OWL, and BESA as the local pruning method and evaluation on the zero-shot tasks are shown Tables 10, 11, 12, and 13 in Appendix.

Table 1: WikiText2 perplexity of pruned LLMs under 50% sparsity ratio using Wanda as the local pruning method.

| Method | Compar. Group | Prune Stra. | Weight Scaling | LLaMA 7B | LLaMA 13B | LLaMA 30B | LLaMA-2 7B | LLaMA-2 13B | LLaMA-3 8B |
|---|---|---|---|---|---|---|---|---|---|
| Dense | - | - | - | 5.67 | 5.09 | 4.10 | 5.11 | 4.57 | 7.46 |
| Centralized | - | - | - | 7.25 | 6.15 | 5.24 | 6.46 | 5.58 | 11.00 |
| Local-only | - | - | - | 7.44 | 6.33 | 5.34 | 6.63 | 5.72 | 11.39 |
| FedPrLLM | Layer | One-shot | ✗ | 7.32 | **6.19** | **5.24** | **6.48** | **5.61** | **11.02** |
| | Row | One-shot | ✗ | **7.30** | 6.20 | 5.25 | **6.48** | **5.61** | **11.02** |
| | Column | One-shot | ✗ | 1524.28 | 9282.09 | 501.88 | 20528.41 | 5309.48 | 311468.53 |
| | Layer | Iterative | ✗ | **7.30** | **6.19** | **5.24** | **6.48** | 5.62 | 11.12 |
| | Row | Iterative | ✗ | **7.30** | 6.20 | **5.24** | **6.48** | **5.61** | 11.11 |
| | Column | Iterative | ✗ | 1822.89 | 6884.15 | 996.57 | 77245.84 | 5430.81 | 189134.78 |
| | Layer | One-shot | ✓ | 7.48 | 6.36 | 5.35 | 6.67 | 5.75 | 11.75 |
| | Row | One-shot | ✓ | 7.47 | 6.36 | 5.35 | 6.67 | 5.75 | 11.75 |
| | Column | One-shot | ✓ | 1708.41 | 10819.42 | 824.50 | 18084.02 | 5914.91 | 276031.34 |
| | Layer | Iterative | ✓ | 7.46 | 6.35 | 5.34 | 6.67 | 5.75 | 11.86 |
| | Row | Iterative | ✓ | 7.46 | 6.35 | 5.34 | 6.67 | 5.74 | 11.87 |
| | Column | Iterative | ✓ | 1985.40 | 6692.91 | 939.62 | 66911.49 | 5268.71 | 41996.95 |

### 4.2.1 WHICH COMPARISON GROUP IS MORE EFFECTIVE?

As discussed above, various comparison groups can be used to select top-k values from the aggregated mask matrix to derive the final mask matrix for pruning the global model, including *layer comparison*, *row comparison*, and *column comparison*. Thus, which comparison group is the most effective?

---

[2]https://github.com/EleutherAI/lm-evaluation-harness

According to the results in Table 1, we observe that layer comparison and row comparison achieve comparable performance, both significantly surpassing column comparison. Results on higher sparsity ratios and other datasets (Tables 6, 7, and 8 in Appendix), using other local pruning methods (Table 10 in Appendix), and results on zero-shot tasks (Table 12 in Appendix) show a similar phenomenon. To investigate why column comparison performs much worse than the others, we noted that the local pruning methods we used adopts row comparison, meaning the local pruning mask matrix $\mathbf{M}_i$ derived from each client is based on row comparison. We hypothesize that this is the reason for the poorer performance of column comparison, as the comparison group used in FedPrLLM conflicts with that of the local pruning method.

Table 2: WikiText2 perplexity of pruned LLMs under 50% sparsity ratio when changing the comparison group for the local pruning method (i.e., Wanda). FedPrLLM adopts one-shot pruning and no weight scaling.

| Local Compar. Group | Method | Compar. Group | LLaMA 7B | LLaMA 13B | LLaMA 30B | LLaMA-2 7B | LLaMA-2 13B | LLaMA-3 8B |
|---|---|---|---|---|---|---|---|---|
| Layer | Centralized | - | 7.94 | 6.57 | 5.47 | 7.38 | 5.92 | 12.04 |
| | Local-only | - | 8.16 | 6.74 | 5.58 | 7.56 | 6.06 | 12.43 |
| | FedPrLLM | Layer | **7.98** | **6.60** | **5.48** | **7.38** | **5.95** | **12.09** |
| | | Row | 31.85 | 10.08 | 11.33 | 39.07 | 124.08 | 17.51 |
| | | Column | 1749.59 | 10183.32 | 541.62 | 25258.16 | 5503.91 | 336255.96 |
| Column | Centralized | - | 8.86 | 7.68 | 5.67 | 10.41 | 6.38 | 83.67 |
| | Local-only | - | 8.86 | 7.68 | 5.67 | 10.41 | 6.38 | 83.67 |
| | FedPrLLM | Layer | **8.86** | **7.68** | **5.67** | **10.41** | **6.38** | **83.67** |
| | | Row | 138.54 | 100.80 | 49.17 | 764.32 | 2580.88 | 400.95 |
| | | Column | **8.86** | **7.68** | **5.67** | **10.41** | **6.38** | **83.67** |

To validate this, we further change the comparison group in the local pruning method (i.e., Wanda (Sun et al., 2024), SparseGPT (Frantar & Alistarh, 2023), OWL (Yin et al., 2024), and BESA (Xu et al., 2024a)) to layer comparison and column comparison to evaluate the performance of the Fed-PrLLM framework with one-shot pruning and no weight scaling. The results on WikiText2 are shown in Table 2, while results for other datasets are presented in Table 9 in Appendix. More results using other local pruning methods and results on the zero-shot tasks are shown in Tables 11 and 13 in Appendix. From these results, we see that when the comparison group in the local pruning method is changed to layer comparison, only the layer comparison used in FedPrLLM performs well, while row comparison performs poorly and column comparison performs even worse. Similarly, when the local pruning method's comparison group is changed to column comparison, only the layer and column comparisons perform normally, while row comparison performance is poor. Note that when the comparison group in the local pruning method is changed to column comparison, it degrades to the magnitude-based pruning method, rendering the performance irrelevant to calibration samples, which results in the performance of Centralized and Local-only being the same (Sun et al., 2024). These results demonstrate our hypothesis that the conflict between the local and server comparison groups leads to worse performance, while the layer comparison used in FerPrLLM consistently achieves good results, regardless of the comparison group used for the local pruning method. The reason for this phenomenon may be due to the mismatch between the local and server comparison groups, which renders the aggregated mask matrix "meaningless". We know that the aggregated mask matrix can be considered a "weight importance matrix" for conducting pruning on the server side. Note that these importance values are only meaningful under the local comparison group and will be meaningless under a mismatched comparison group. Therefore, when the comparison group used on the server mismatches the local group (e.g., local-row and server-column), the aggregated mask matrix will be meaningless and cannot be used to determine which weights are important, leading to poor pruning results. However, the layer comparison used on the server can avoid this issue since the comparisons within the whole layer will also take the local comparison group into consideration. Thus, regardless of the local comparison group used on the client side, the layer comparison used on the server can achieve good results. Therefore, we conclude that:

> **Takeaway 1:** Layer comparison is simple yet effective.

### 4.2.2 SHOULD WE SCALE THE MODEL WEIGHTS OF THE RETAINED PARAMETERS?

The aggregated mask matrix $\hat{\mathbf{M}}$ indicates the number of clients that wish to prune a parameter, which allows it to be used for scaling the model weights of the retained parameters to $\frac{(m-\hat{\mathbf{M}})}{m}$. This approach corresponds to locally pruning the model and then sharing the pruned model with the server, which aggregates them using the FedAvg algorithm (McMahan et al., 2017). However, will weight scaling be beneficial for the federated pruning of LLMs?

From the results in Table 1, we observe that the performance with weight scaling is worse than that without weight scaling across all comparison groups and pruning strategies. Results on higher sparsity ratios and more datasets (Tables 6, 7, and 8 in Appendix), using other local pruning methods (Table 10 in Appendix), and results on zero-shot tasks (Table 12 in Appendix) show a similar phenomenon. It indicates that scaling weights offers no benefit and may even worsen performance. This may be due to the fact that locally pruned models do not perform well, and applying the FedAvg algorithm (McMahan et al., 2017) to aggregate these pruned model weights leads to subpar performance. Therefore, we conclude that:

> **Takeaway 2:** Scaling weights performs worse than expected.

### 4.2.3 DOES ITERATIVE PRUNING OUTPERFORM ONE-SHOT PRUNING?

Since LLMs are usually pruned *layer-by-layer* recursively (Frantar & Alistarh, 2023; Sun et al., 2024; Zhang et al., 2024b), federated pruning for LLMs can be naturally categorized into two types: *one-shot pruning* and *iterative pruning*. Given the significant communication costs associated with iterative pruning, will it outperform one-shot pruning?

Table 3: Communication cost for one-shot and iterative pruning. The unit is the number of parameters and "B" denotes billions.

|  | LLaMA-7B | LLaMA-13B | LLaMA-30B | LLaMA-2-7B | LLaMA-2-13B | LLaMA-3-8B |
|---|---|---|---|---|---|---|
| one-shot pruning | 6.476B | 12.688B | 32.102B | 6.476B | 12.688B | 6.979B |
| iterative pruning | 12.952B | 25.376B | 64.204B | 12.952B | 25.376B | 13.958B |

The comparison results are provided in Table 1, More results on higher sparsity ratios and other datasets are shown in Tables 6, 7, and 8 in Appendix. Results using other local pruning methods are shown in Table 10 in Appendix, and results on zero-shot tasks are shown in Table 12 in Appendix. These results indicate that the performance of iterative pruning and one-shot pruning is comparable, regardless of the comparison groups and pruning strategies. However, since iterative pruning introduces significant communication costs (Table 3) without any performance improvement (see Section D in Appendix for more comparisons in terms of efficiency), we conclude that:

> **Takeaway 3:** Iterative pruning offers no benefit.

### 4.3 EXTENSION TO NON-IID SCENARIOS

To validate the generalizability of our findings, we further conduct experiments under non-IID conditions. Specifically, we extract 8 samples from the training data of WikiText2 (Merity et al., 2017), C4 (Raffel et al., 2020), and PTB (Marcus et al., 1994) to form a global calibration dataset (i.e., 24 samples in total). We then use the Dirichlet distribution with a concentration parameter of $\alpha = 5$ to split the global calibration dataset into 12 non-IID local calibration datasets, each assigned to one client (i.e., 2 samples per client). We choose Wanda as the local pruning method and use LLaMA-7B to conduct experiments with 50% sparsity pruning. The experimental results under non-IID conditions are shown in Tables 4 and 5. As shown in these results, our proposed "Best Recipe"—using one-shot pruning, layer-wise comparison, and no weight scaling—consistently outperforms other configurations under the non-IID scenario, confirming that our findings are generalizable.

Table 4: Perplexity (WikiText2 / C4 / PTB) of pruned LLMs under 50% sparsity ratio using Wanda as the local pruning method under non-IID conditions.

| Method | Compar. Group | Prune Stra. | Weight Scaling | LLaMA-7B |
|---|---|---|---|---|
| Centralized | - | - | - | 7.06 / 9.27 / 65.72 |
| Local-only | - | - | - | 7.16 / 9.42 / 71.54 |
| FedPrLLM | Layer | One-shot | ✗ | **7.06 / 9.30** / 67.54 |
| | Row | One-shot | ✗ | **7.06 / 9.30 / 67.28** |
| | Column | One-shot | ✗ | 2923.46 / 1813.31 / 6736.30 |
| | Layer | Iterative | ✗ | **7.06** / 9.31 / 68.09 |
| | Row | Iterative | ✗ | **7.06 / 9.30** / 67.34 |
| | Column | Iterative | ✗ | 3219.96 / 2294.87 / 6812.14 |
| | Layer | One-shot | ✓ | 7.17 / 9.47 / 72.33 |
| | Row | One-shot | ✓ | 7.17 / 9.47 / 72.16 |
| | Column | One-shot | ✓ | 2723.30 / 1554.46 / 6364.29 |
| | Layer | Iterative | ✓ | 7.17 / 9.48 / 73.40 |
| | Row | Iterative | ✓ | 7.17 / 9.48 / 72.92 |
| | Column | Iterative | ✓ | 3182.52 / 1795.12 / 5808.61 |

Table 5: Perplexity (WikiText2 / C4 / PTB) of pruned LLMs under 50% sparsity ratio when changing the comparison group for the local pruning method (i.e., Wanda) under non-IID conditions. FedPrLLM adopts one-shot pruning and no weight scaling.

| Local Compar. Group | Method | Compar. Group | LLaMA-7B |
|---|---|---|---|
| Layer | Centralized | - | 7.67 / 10.07 / 83.20 |
| | Local-only | - | 7.76 / 10.26 / 85.16 |
| | FedPrLLM | Layer | **7.62 / 10.10 / 81.70** |
| | | Row | 43.54 / 46.29 / 348.41 |
| | | Column | 2324.40 / 1434.18 / 6026.79 |
| Column | Centralized | - | 8.86 / 14.10 / 108.37 |
| | Local-only | - | 8.86 / 14.10 / 108.37 |
| | FedPrLLM | Layer | **8.86 / 14.10 / 108.37** |
| | | Row | 138.54 / 155.15 / 1060.99 |
| | | Column | **8.86 / 14.10 / 108.37** |

## 4.4 SENSITIVITY ANALYSIS

In this section, we conduct sensitivity analyses on the number of clients and calibration samples in FedPrLLM to better understand its effectiveness in pruning LLMs within a federated scenario. We utilize Wanda as the local pruning method and use FedPrLLM, which employs layer comparison, one-shot pruning, and no weight scaling, to conduct the analysis under a 50% sparsity ratio.

It is worth noting that the number of clients influences the performance of FL algorithms (Guo et al., 2025b;c). In this section, we investigate the effect of client numbers on the federated pruning of LLMs. We use a total of 128 calibration samples and vary the number of clients from 64 to 2, resulting in an increase in the calibration samples allocated to each client. Specifically, when the number of clients is 64, each client has only 2 calibration samples; when the number of clients is reduced to 2, each client has 64 calibration samples. The experimental results are shown in Figure 2. From this figure, we observe that FedPrLLM consistently outperforms Local-only pruning across various numbers of clients, demonstrating the effectiveness of the federated pruning algorithm.

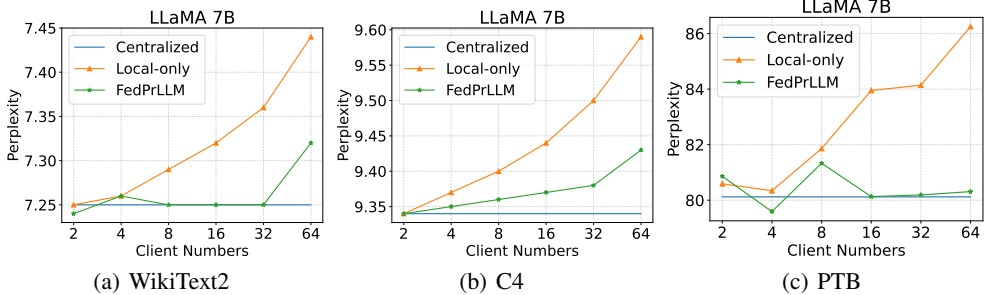

| (a) WikiText2 | (b) C4 | (c) PTB |
|---|---|---|

Figure 2: The effect of different client numbers on federated pruning LLMs.

We further investigate the impact of pruning LLMs in a federated scenario with varying numbers of calibration samples, as shown in Figure 3. Specifically, we change the total number of calibration samples from 128 to 4 while keeping the number of clients equal to half of that. As shown in Figure 3, we observe that with different numbers of calibration samples, FedPrLLM consistently outperforms Local-only pruning, which again shows the effectiveness of the federated pruning method.

## 4.5 PRIVACY AND LEAKAGE ANALYSIS

In this section, we conduct a detailed privacy analysis to formally and empirically assess the privacy leakage of our framework for the LLaMA-7B model, covering both theoretical limits and practical attack simulations.

To measure maximum information leakage, we conduct an information entropy analysis revealing that a binary mask at 50% sparsity holds only 1.0 bit of information, compared to 13.75 bits for standard Float16 model weights, indicating a 92.7% reduction in information. This substantial reduction

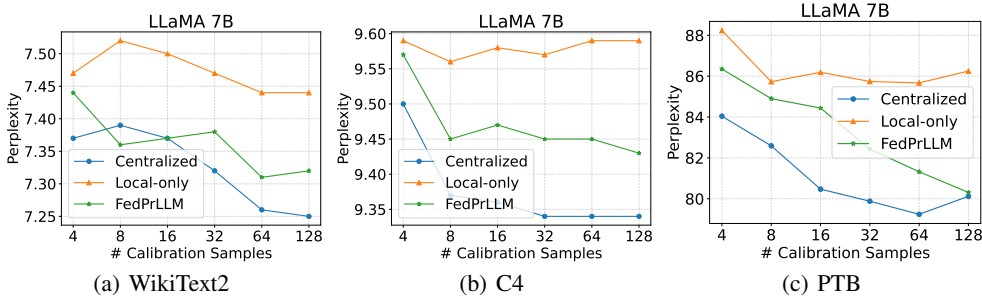

Figure 3: The effect of the number of calibration samples on federated pruning LLMs.

enhances security by making attacks more challenging. We further investigate our mask-sharing method through practical experiments, finding that masks generated with Wanda from randomly seeded calibration data are over 95% identical, which suggests they are primarily determined by the public pre-trained model, thus separating shared information from private data. Our Differential Privacy (Dwork, 2006) sensitivity analysis shows that altering a single dataset sample results in only a 4.96% change in the mask, providing strong privacy protection equivalent to a formal privacy budget of $\epsilon \approx 0.05$ without added noise. We also simulate targeted attacks to assess privacy leakage, including Membership Inference Attacks (Shokri et al., 2017), where the difference in masks—with and without a target sample—yields only a 3.23% Hamming distance, making it difficult to distinguish between signals and noise. Finally, in Gradient Inversion Attacks (Zhu et al., 2019; Fredrikson et al., 2015), the attacker also fails to reconstruct original training data, recovering less than 2% of tokens and generating nonsensical text. See Section C in Appendix for more details.

Therefore, by sharing only low-information binary masks, our framework fundamentally reduces privacy risks and offers strong, practical privacy protection.

## 5 RELATED WORK

There is one work that attempts to conduct LLM pruning in an FL scenario, i.e., FedSpaLLM (Bai et al., 2024). It enables clients to collaboratively prune an LLM by introducing an $\ell_0$-norm aggregation function, an adaptive mask expansion technique, and a layer sampling strategy. While FedSpaLLM proposes a specific and novel algorithm for federated LLM pruning, our paper provides the first systematic and comprehensive empirical study of the fundamental design space of federated LLM pruning. Our primary goal is not to introduce another single algorithm, but to establish a set of generalizable "best practices" and a "recipe" that can guide future research and applications in this domain. Moreover, FedSpaLLM's core operation can be mapped to a specific configuration within our comprehensive FedPrLLM framework. Specifically, it enables clients to locally prune their models based on private data and send the pruned models to the server for aggregation. The server averages the pruned models using the FedAvg algorithm (McMahan et al., 2017) and prunes the model to satisfy the predefined sparsity rate based on an aggregated mask matrix. This method can be viewed as a specific case within our FedPrLLM framework, i.e., iterative pruning with weight scaling. However, our extensive evaluations reveal that this approach is not optimal.

## 6 CONCLUSION

In this work, we introduce **FedPrLLM**, a comprehensive federated pruning framework designed for the privacy-preserving compression of LLMs, incorporating various possibilities for integrating FL with LLM pruning. To identify the optimal operation within this framework, we invested thousands of GPU hours exploring these possibilities, including different comparison groups, pruning strategies, and the decision to scale weights. Our extensive evaluation reveals that one-shot pruning with layer comparison and no weight scaling is the optimal choice within the FedPrLLM framework. We hope our work will help guide future efforts in pruning LLMs in privacy-sensitive fields.

**Future Work.** This work currently focuses on unstructured pruning of LLMs in a federated scenario. Future work could explore structured pruning within the FedPrLLM framework, which may be more suitable for certain real-world applications due to its hardware efficiency.

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

## A    THE RATIONALE OF VOTING MECHANISMS

In this section, we provide theoretical analysis to demonstrate the rationale behind the voting mechanism for deriving the final pruning mask on the server side. Let $\{\mathbf{M}_1, \ldots, \mathbf{M}_m\}$ be $m$ independent $d \times r$ binary mask matrices (with 50% sparsity) where each matrix satisfies:

$$\sum_{p=1}^{d}\sum_{q=1}^{r} \mathbf{M}_i[p, q] = \frac{dr}{2}. \tag{2}$$

The voting mechanism procedure produces $\mathbf{M}$ via: 1) Element-wise sum: $\hat{\mathbf{M}} = \sum_{i=1}^{m} \mathbf{M}_i$; 2) Set the largest $\frac{dr}{2}$ entries in $\hat{\mathbf{M}}$ to 1, others to 0.

Let $\mathbf{M}^*$ be the optimal mask defined by:

$$\mathbf{M}^*[p, q] = \mathbb{I}\left(p_{pq} \geq \tau^*\right), \tag{3}$$

where $p_{pq} = P(\mathbf{M}_i[p, q] = 1)$ and $\tau^*$ is chosen such that $\sum_{p,q} \mathbf{M}^*[p, q] = \frac{dr}{2}$.

Then, the error between $\mathbf{M}$ and $\mathbf{M}^*$ can be defined as:

$$\epsilon = \frac{1}{dr}\sum_{p=1}^{d}\sum_{q=1}^{r} \mathbb{I}\left(\mathbf{M}[p, q] \neq \mathbf{M}^*[p, q]\right). \tag{4}$$

There are two situations for $\mathbf{M}^*[p, q]$: 1 or 0.

**Case 1:** $\mathbf{M}^*[p, q] = 1$ (i.e., $p_{pq} \geq \tau^*$). In this case, $\mathbf{M}[p, q] = 0$ implies $\frac{\hat{\mathbf{M}}[p,q]}{m} < \tau^*$. Thus:

$$p_{pq} - \frac{\hat{\mathbf{M}}[p, q]}{m} > p_{pq} - \tau^* = \delta_{pq} \quad (\text{since } \delta_{pq} = |p_{pq} - \tau^*| = p_{pq} - \tau^*), \tag{5}$$

which simplifies to:

$$|\frac{\hat{\mathbf{M}}[p, q]}{m} - p_{pq}| > \delta_{pq} \tag{6}$$

**Case 2:** $\mathbf{M}^*[p, q] = 0$ (i.e., $p_{pq} < \tau^*$). In this case, $\mathbf{M}[p, q] = 1$ implies $\frac{\hat{\mathbf{M}}[p,q]}{m} \geq \tau^*$. Thus:

$$\frac{\hat{\mathbf{M}}[p, q]}{m} - p_{pq} \geq \tau^* - p_{pq} = \delta_{pq} \quad (\text{since } \delta_{pq} = \tau^* - p_{pq}), \tag{7}$$

which simplifies to:

$$\left|\frac{\hat{\mathbf{M}}[p, q]}{m} - p_{pq}\right| \geq \delta_{pq} \tag{8}$$

Let: **Event** $A$: $\mathbf{M}[p, q] \neq \mathbf{M}^*[p, q]$; **Event** $B$: $\left|\frac{\hat{\mathbf{M}}[p,q]}{m} - p_{pq}\right| \geq \frac{\delta_{pq}}{2}$. Then $A \subseteq B$, and we have:

$$P\left(\mathbf{M}[p, q] \neq \mathbf{M}^*[p, q]\right) \leq P\left(\left|\frac{\hat{\mathbf{M}}[p, q]}{m} - p_{pq}\right| \geq \frac{\delta_{pq}}{2}\right) \leq 2\exp\left(-\frac{m\delta_{pq}^2}{2}\right), \tag{9}$$

This implies:

$$\mathbb{E}[\epsilon] \leq \frac{2}{dr}\sum_{p,q} \exp\left(-\frac{m\delta_{pq}^2}{2}\right). \tag{10}$$

This shows that the error between $\mathbf{M}$ (which is obtained by voting) and $\mathbf{M}^*$ is bounded by some value, which demonstrates the rationale behind the voting mechanism.

# B ADDITIONAL EXPERIMENTAL RESULTS

## B.1 MORE RESULTS UNDER HIGHER SPARSITY RATIOS

The experimental results using Wanda as the local pruning method for higher sparsity ratios (i.e., 60% and 70%) are shown in Tables 6, 7, and 8.

Table 6: WikiText2 perplexity of pruned LLaMA, LLaMA-2, and LLaMA-3 models using Wanda as the local pruning method.

| Sparsity | Method | Compar. Group | Prune Stra. | Weight Scaling | LLaMA 7B | LLaMA 13B | LLaMA 30B | LLaMA-2 7B | LLaMA-2 13B | LLaMA-3 8B |
|---|---|---|---|---|---|---|---|---|---|---|
| 0% | Dense | - | - | - | 5.67 | 5.09 | 4.10 | 5.11 | 4.57 | 7.46 |
| 50% | Centralized | - | - | - | 7.25 | 6.15 | 5.24 | 6.46 | 5.58 | 11.00 |
| | Local-only | - | - | - | 7.44 | 6.33 | 5.34 | 6.63 | 5.72 | 11.39 |
| | FedPrLLM | Layer | One-shot | ✗ | 7.32 | 6.19 | 5.24 | 6.48 | 5.61 | 11.02 |
| | | Row | One-shot | ✗ | 7.30 | 6.20 | 5.25 | 6.48 | 5.61 | 11.02 |
| | | Column | One-shot | ✗ | 1524.28 | 9282.09 | 501.88 | 20528.41 | 5309.48 | 311468.53 |
| | | Layer | Iterative | ✗ | 7.30 | 6.19 | 5.24 | 6.48 | 5.62 | 11.12 |
| | | Row | Iterative | ✗ | 7.30 | 6.20 | 5.24 | 6.48 | 5.61 | 11.11 |
| | | Column | Iterative | ✗ | 1822.89 | 6884.15 | 996.57 | 77245.84 | 5430.81 | 189134.78 |
| | | Layer | One-shot | ✓ | 7.48 | 6.36 | 5.35 | 6.67 | 5.75 | 11.75 |
| | | Row | One-shot | ✓ | 7.47 | 6.36 | 5.35 | 6.67 | 5.75 | 11.75 |
| | | Column | One-shot | ✓ | 1708.41 | 10819.42 | 824.5 | 18084.02 | 5914.91 | 276031.34 |
| | | Layer | Iterative | ✓ | 7.46 | 6.35 | 5.34 | 6.67 | 5.75 | 11.86 |
| | | Row | Iterative | ✓ | 7.46 | 6.35 | 5.34 | 6.67 | 5.74 | 11.87 |
| | | Column | Iterative | ✓ | 1985.40 | 6692.91 | 939.62 | 66911.49 | 5268.71 | 41996.95 |
| 60% | Centralized | - | - | - | 10.71 | 8.74 | 6.55 | 10.03 | 7.92 | 25.81 |
| | Local-only | - | - | - | 11.70 | 9.38 | 6.96 | 10.84 | 8.55 | 27.47 |
| | FedPrLLM | Layer | One-shot | ✗ | 10.76 | 8.80 | 6.65 | 10.08 | 8.01 | 25.48 |
| | | Row | One-shot | ✗ | 10.77 | 8.80 | 6.64 | 10.08 | 8.03 | 25.64 |
| | | Column | One-shot | ✗ | 2861.56 | 11190.34 | 1047.94 | 14737.65 | 5385.33 | 382319.37 |
| | | Layer | Iterative | ✗ | 10.87 | 8.88 | 6.65 | 10.17 | 8.05 | 26.21 |
| | | Row | Iterative | ✗ | 10.85 | 8.90 | 6.64 | 10.18 | 8.05 | 25.98 |
| | | Column | Iterative | ✗ | 3154.68 | 7824.46 | 2250.97 | 18849.20 | 6556.50 | 65475.84 |
| | | Layer | One-shot | ✓ | 12.14 | 9.77 | 7.10 | 11.53 | 8.98 | 30.34 |
| | | Row | One-shot | ✓ | 12.16 | 9.77 | 7.09 | 11.53 | 9.00 | 30.44 |
| | | Column | One-shot | ✓ | 3785.85 | 17163.16 | 1770.89 | 15180.33 | 5401.19 | 608169.33 |
| | | Layer | Iterative | ✓ | 12.27 | 9.85 | 7.12 | 11.90 | 9.07 | 30.94 |
| | | Row | Iterative | ✓ | 12.24 | 9.86 | 7.13 | 11.87 | 9.06 | 31.08 |
| | | Column | Iterative | ✓ | 2189.53 | 6032.71 | 2626.57 | 16081.73 | 6227.41 | 165510.73 |
| 70% | Centralized | - | - | - | 87.42 | 53.48 | 17.30 | 72.38 | 45.94 | 92.20 |
| | Local-only | - | - | - | 104.15 | 67.13 | 23.29 | 80.39 | 51.79 | 108.35 |
| | FedPrLLM | Layer | One-shot | ✗ | 83.12 | 55.92 | 18.73 | 70.92 | 44.98 | 102.88 |
| | | Row | One-shot | ✗ | 81.97 | 56.99 | 18.67 | 70.61 | 44.66 | 102.13 |
| | | Column | One-shot | ✗ | 17281.43 | 13045.16 | 2670.43 | 31238.51 | 12206.74 | 458666.00 |
| | | Layer | Iterative | ✗ | 89.25 | 55.48 | 18.65 | 79.27 | 45.89 | 100.37 |
| | | Row | Iterative | ✗ | 92.29 | 57.18 | 18.23 | 72.60 | 45.68 | 93.13 |
| | | Column | Iterative | ✗ | 19791.05 | 10323.63 | 3935.54 | 23090.20 | 7857.41 | 355916.56 |
| | | Layer | One-shot | ✓ | 136.50 | 94.90 | 31.62 | 93.89 | 64.34 | 123.92 |
| | | Row | One-shot | ✓ | 136.09 | 95.86 | 31.48 | 93.36 | 63.98 | 124.65 |
| | | Column | One-shot | ✓ | 20505.56 | 11695.06 | 3032.65 | 31485.38 | 10875.86 | 831352.18 |
| | | Layer | Iterative | ✓ | 174.95 | 102.78 | 31.12 | 94.49 | 62.07 | 116.97 |
| | | Row | Iterative | ✓ | 182.73 | 99.32 | 30.87 | 96.37 | 62.51 | 120.19 |
| | | Column | Iterative | ✓ | 8607.36 | 11707.00 | 3145.32 | 36254172.00 | 9604.48 | 1034635.56 |

Table 7: C4 perplexity of pruned LLaMA, LLaMA-2, and LLaMA-3 models using Wanda as the local pruning method.

| Sparsity | Method | Compar. Group | Prune Stra. | Weight Scaling | LLaMA 7B | LLaMA 13B | LLaMA 30B | LLaMA-2 7B | LLaMA-2 13B | LLaMA-3 8B |
|---|---|---|---|---|---|---|---|---|---|---|
| 0% | Dense | - | - | - | 7.34 | 6.79 | 6.12 | 7.03 | 6.51 | 12.34 |
| 50% | Centralized | - | - | - | 9.34 | 8.14 | 7.28 | 8.94 | 8.03 | 18.38 |
| | Local-only | - | - | - | 9.59 | 8.37 | 7.52 | 9.16 | 8.31 | 18.92 |
| | FedPrLLM | Layer | One-shot | ✗ | 9.43 | 8.22 | 7.39 | 9.01 | 8.18 | 18.32 |
| | | Row | One-shot | ✗ | 9.43 | 8.22 | 7.39 | 9.01 | 8.19 | 18.32 |
| | | Column | One-shot | ✗ | 893.05 | 10616.94 | 612.27 | 9631.37 | 5075.92 | 200257.70 |
| | | Layer | Iterative | ✗ | 9.44 | 8.22 | 7.39 | 9.01 | 8.19 | 18.43 |
| | | Row | Iterative | ✗ | 9.44 | 8.22 | 7.39 | 9.02 | 8.18 | 18.38 |
| | | Column | Iterative | ✗ | 1050.26 | 8567.66 | 779.01 | 11658.80 | 4804.46 | 112192.42 |
| | | Layer | One-shot | ✓ | 9.64 | 8.40 | 7.57 | 9.21 | 8.39 | 19.45 |
| | | Row | One-shot | ✓ | 9.64 | 8.40 | 7.57 | 9.21 | 8.39 | 19.45 |
| | | Column | One-shot | ✓ | 887.34 | 13744.66 | 895.18 | 11440.51 | 5189.73 | 90476.94 |
| | | Layer | Iterative | ✓ | 9.64 | 8.41 | 7.57 | 9.22 | 8.39 | 19.58 |
| | | Row | Iterative | ✓ | 9.65 | 8.41 | 7.57 | 9.22 | 8.39 | 19.60 |
| | | Column | Iterative | ✓ | 1242.31 | 6860.69 | 724.28 | 10355.87 | 4657.88 | 44469.52 |
| 60% | Centralized | - | - | - | 13.72 | 11.22 | 9.16 | 13.64 | 11.39 | 43.02 |
| | Local-only | - | - | - | 14.69 | 11.91 | 9.58 | 14.68 | 12.17 | 45.25 |
| | FedPrLLM | Layer | One-shot | ✗ | 13.80 | 11.23 | 9.29 | 13.77 | 11.40 | 42.61 |
| | | Row | One-shot | ✗ | 15.26 | 12.24 | 9.79 | 15.60 | 12.75 | 50.37 |
| | | Column | One-shot | ✗ | 2149.09 | 11488.68 | 993.56 | 12252.16 | 4606.43 | 837570.62 |
| | | Layer | Iterative | ✗ | 13.92 | 11.37 | 9.32 | 13.84 | 11.52 | 44.24 |
| | | Row | Iterative | ✗ | 13.86 | 11.38 | 9.30 | 13.85 | 11.53 | 43.77 |
| | | Column | Iterative | ✗ | 2981.52 | 10375.02 | 1752.73 | 16673.62 | 5289.35 | 62234.32 |
| | | Layer | One-shot | ✓ | 15.24 | 12.24 | 9.80 | 15.61 | 12.74 | 50.28 |
| | | Row | One-shot | ✓ | 15.26 | 12.24 | 9.79 | 15.60 | 12.75 | 50.37 |
| | | Column | One-shot | ✓ | 3336.72 | 19430.46 | 1520.32 | 14613.11 | 4547.54 | 622715.25 |
| | | Layer | Iterative | ✓ | 15.46 | 12.54 | 9.86 | 16.15 | 13.01 | 51.47 |
| | | Row | Iterative | ✓ | 15.42 | 12.54 | 9.87 | 16.10 | 13.01 | 51.48 |
| | | Column | Iterative | ✓ | 1825.82 | 6669.63 | 1865.50 | 16167.12 | 5057.57 | 145341.28 |
| 70% | Centralized | - | - | - | 85.84 | 53.35 | 18.80 | 84.16 | 58.56 | 136.66 |
| | Local-only | - | - | - | 96.47 | 63.61 | 22.48 | 82.96 | 67.09 | 161.86 |
| | FedPrLLM | Layer | One-shot | ✗ | 81.95 | 52.55 | 19.24 | 81.40 | 59.87 | 158.08 |
| | | Row | One-shot | ✗ | 82.02 | 53.51 | 19.22 | 81.59 | 59.97 | 157.87 |
| | | Column | One-shot | ✗ | 15276.62 | 14041.01 | 2059.83 | 39339.21 | 11306.11 | 398674.93 |
| | | Layer | Iterative | ✗ | 83.52 | 57.22 | 19.15 | 92.51 | 60.46 | 162.29 |
| | | Row | Iterative | ✗ | 86.77 | 55.98 | 19.20 | 84.99 | 60.86 | 144.71 |
| | | Column | Iterative | ✗ | 18149.76 | 13537.18 | 2874.83 | 21704.32 | 7166.78 | 346598.5 |
| | | Layer | One-shot | ✓ | 116.61 | 77.99 | 26.30 | 104.86 | 79.82 | 184.11 |
| | | Row | One-shot | ✓ | 117.29 | 78.84 | 26.29 | 104.51 | 79.76 | 184.11 |
| | | Column | One-shot | ✓ | 19380.0 | 10934.98 | 2336.68 | 32034.07 | 11360.57 | 345798.53 |
| | | Layer | Iterative | ✓ | 142.08 | 85.91 | 27.17 | 103.02 | 79.25 | 177.97 |
| | | Row | Iterative | ✓ | 145.15 | 84.68 | 27.00 | 102.61 | 79.41 | 182.36 |
| | | Column | Iterative | ✓ | 7664.62 | 15985.50 | 2685.76 | 27805842.0 | 8041.09 | 1031318.56 |

Table 8: PTB perplexity of pruned LLaMA, LLaMA-2, and LLaMA-3 models using Wanda as the local pruning method.

| Sparsity | Method | Compar. Group | Prune Stra. | Weight Scaling | LLaMA 7B | LLaMA 13B | LLaMA 30B | LLaMA-2 7B | LLaMA-2 13B | LLaMA-3 8B |
|---|---|---|---|---|---|---|---|---|---|---|
| 0% | Dense | - | - | - | 41.15 | 28.09 | 23.51 | 50.20 | 56.51 | 13.30 |
| 50% | Centralized | - | - | - | 80.12 | 36.41 | 26.64 | 96.99 | 86.83 | 20.69 |
| | Local-only | - | - | - | 86.25 | 37.57 | 27.13 | 108.66 | 91.92 | 21.43 |
| | FedPrLLM | Layer | One-shot | ✗ | 80.31 | 36.57 | 26.69 | 102.71 | 88.26 | 20.56 |
| | | Row | One-shot | ✗ | 80.71 | 36.61 | 26.64 | 101.85 | 88.31 | 20.55 |
| | | Column | One-shot | ✗ | 4463.92 | 22138.56 | 713.56 | 14256.86 | 7392.64 | 407313.84 |
| | | Layer | Iterative | ✗ | 81.22 | 36.54 | 26.68 | 102.72 | 88.38 | 20.55 |
| | | Row | Iterative | ✗ | 81.26 | 36.55 | 26.64 | 103.66 | 88.94 | 20.60 |
| | | Column | Iterative | ✗ | 4061.96 | 17610.52 | 1158.75 | 13401.63 | 6941.72 | 168643.04 |
| | | Layer | One-shot | ✓ | 87.97 | 37.70 | 27.27 | 112.52 | 92.90 | 22.21 |
| | | Row | One-shot | ✓ | 88.35 | 37.72 | 27.25 | 112.17 | 93.07 | 22.21 |
| | | Column | One-shot | ✓ | 4557.48 | 29140.28 | 982.59 | 12021.08 | 7801.23 | 264723.12 |
| | | Layer | Iterative | ✓ | 87.28 | 37.69 | 27.27 | 112.95 | 92.58 | 22.39 |
| | | Row | Iterative | ✓ | 87.61 | 37.60 | 27.27 | 113.32 | 92.61 | 22.41 |
| | | Column | Iterative | ✓ | 6929.83 | 15189.83 | 1178.40 | 10208.03 | 5220.64 | 39172.53 |
| 60% | Centralized | - | - | - | 193.10 | 71.66 | 34.94 | 363.71 | 220.81 | 52.42 |
| | Local-only | - | - | - | 208.48 | 82.24 | 37.27 | 409.47 | 271.49 | 55.39 |
| | FedPrLLM | Layer | One-shot | ✗ | 187.00 | 74.66 | 35.38 | 339.79 | 241.14 | 52.61 |
| | | Row | One-shot | ✗ | 186.10 | 74.64 | 35.47 | 337.69 | 242.96 | 52.61 |
| | | Column | One-shot | ✗ | 5604.92 | 31222.37 | 1338.25 | 28046.95 | 7553.32 | 322022.84 |
| | | Layer | Iterative | ✗ | 191.22 | 72.90 | 35.83 | 368.87 | 237.45 | 53.78 |
| | | Row | Iterative | ✗ | 190.60 | 73.74 | 35.77 | 367.56 | 235.51 | 53.25 |
| | | Column | Iterative | ✗ | 6785.79 | 13234.02 | 1903.66 | 24022.75 | 8125.57 | 46139.19 |
| | | Layer | One-shot | ✓ | 216.09 | 91.63 | 38.22 | 429.58 | 293.11 | 60.49 |
| | | Row | One-shot | ✓ | 215.50 | 91.60 | 38.25 | 428.87 | 294.44 | 60.48 |
| | | Column | One-shot | ✓ | 7600.58 | 41079.65 | 1910.36 | 18249.40 | 7601.34 | 416094.71 |
| | | Layer | Iterative | ✓ | 220.22 | 90.60 | 38.79 | 427.12 | 283.34 | 61.25 |
| | | Row | Iterative | ✓ | 220.16 | 90.58 | 38.74 | 428.36 | 282.20 | 61.55 |
| | | Column | Iterative | ✓ | 4242.84 | 11345.68 | 2133.62 | 29512.89 | 7113.24 | 133467.18 |
| 70% | Centralized | - | - | - | 698.79 | 299.42 | 110.70 | 1902.56 | 735.73 | 131.13 |
| | Local-only | - | - | - | 782.42 | 412.24 | 144.90 | 1780.26 | 863.50 | 152.97 |
| | FedPrLLM | Layer | One-shot | ✗ | 737.07 | 366.28 | 120.33 | 1521.25 | 793.55 | 156.63 |
| | | Row | One-shot | ✗ | 718.37 | 369.65 | 118.24 | 1557.08 | 792.08 | 154.72 |
| | | Column | One-shot | ✗ | 18649.81 | 18136.88 | 3180.23 | 49646.82 | 12010.97 | 466632.84 |
| | | Layer | Iterative | ✗ | 721.31 | 355.21 | 113.31 | 1675.79 | 775.69 | 146.27 |
| | | Row | Iterative | ✗ | 734.43 | 349.63 | 113.65 | 1757.10 | 767.13 | 133.92 |
| | | Column | Iterative | ✗ | 28179.23 | 17249.42 | 3967.48 | 29254.5 | 10233.18 | 314505.62 |
| | | Layer | One-shot | ✓ | 839.42 | 484.11 | 188.18 | 1633.85 | 890.27 | 174.11 |
| | | Row | One-shot | ✓ | 830.33 | 483.58 | 187.11 | 1641.92 | 891.27 | 172.74 |
| | | Column | One-shot | ✓ | 26556.95 | 21627.29 | 3383.87 | 54429.17 | 14951.70 | 239612.84 |
| | | Layer | Iterative | ✓ | 887.36 | 469.70 | 173.86 | 1789.42 | 858.48 | 162.24 |
| | | Row | Iterative | ✓ | 896.85 | 454.31 | 172.48 | 1740.04 | 879.50 | 168.51 |
| | | Column | Iterative | ✓ | 8660.95 | 18472.69 | 3246.05 | 11427895.00 | 8037.55 | 738685.56 |

## B.2 MORE RESULTS ON THE COMPARISON GROUP FOR LOCAL PRUNING

The results of changing the comparison group for the local pruning method (i.e., Wanda) are shown in Table 9.

Table 9: Perplexity of pruned LLMs under 50% sparsity ratio when changing the comparison group for the local pruning method (i.e., Wanda). FedPrLLM adopts one-shot pruning and no weight scaling.

| Local Compar. Group | Dataset | Method | Compar. Group | LLaMA 7B | LLaMA 13B | LLaMA 30B | LLaMA-2 7B | LLaMA-2 13B | LLaMA-3 8B |
|---|---|---|---|---|---|---|---|---|---|
| Layer | WikiText2 | Centralized | - | 7.94 | 6.57 | 5.47 | 7.38 | 5.92 | 12.04 |
| | | Local-only | - | 8.16 | 6.74 | 5.58 | 7.56 | 6.06 | 12.43 |
| | | FedPrLLM | Layer | 7.98 | 6.60 | 5.48 | 7.38 | 5.95 | 12.09 |
| | | | Row | 31.85 | 10.08 | 11.33 | 39.07 | 124.08 | 17.51 |
| | | | Column | 1749.59 | 10183.32 | 541.62 | 25258.16 | 5503.91 | 336255.96 |
| | C4 | Centralized | - | 10.28 | 8.63 | 7.59 | 10.24 | 8.49 | 19.18 |
| | | Local-only | - | 10.56 | 8.90 | 7.86 | 10.52 | 8.76 | 19.64 |
| | | FedPrLLM | Layer | 10.34 | 8.71 | 7.72 | 10.32 | 8.63 | 19.09 |
| | | | Row | 34.90 | 12.35 | 12.75 | 29.79 | 207.57 | 28.05 |
| | | | Column | 975.75 | 12605.58 | 553.85 | 13950.23 | 4899.58 | 129415.62 |
| | PTB | Centralized | - | 92.84 | 43.47 | 27.25 | 306.71 | 119.17 | 23.14 |
| | | Local-only | - | 99.13 | 45.34 | 27.87 | 338.70 | 136.88 | 23.69 |
| | | FedPrLLM | Layer | 91.99 | 43.59 | 27.25 | 305.79 | 124.27 | 22.85 |
| | | | Row | 284.19 | 109.14 | 110.46 | 1886.94 | 480.24 | 44.71 |
| | | | Column | 3976.21 | 28144.48 | 711.16 | 14131.82 | 7134.88 | 293147.84 |
| Column | WikiText2 | Centralized | - | 8.86 | 7.68 | 5.67 | 10.41 | 6.38 | 83.67 |
| | | Local-only | - | 8.86 | 7.68 | 5.67 | 10.41 | 6.38 | 83.67 |
| | | FedPrLLM | Layer | 8.86 | 7.68 | 5.67 | 10.41 | 6.38 | 83.67 |
| | | | Row | 138.54 | 100.80 | 49.17 | 764.32 | 2580.88 | 400.95 |
| | | | Column | 8.86 | 7.68 | 5.67 | 10.41 | 6.38 | 83.67 |
| | C4 | Centralized | - | 14.10 | 11.20 | 8.06 | 17.90 | 9.57 | 30.88 |
| | | Local-only | - | 14.10 | 11.20 | 8.06 | 17.90 | 9.57 | 30.88 |
| | | FedPrLLM | Layer | 14.10 | 11.20 | 8.06 | 17.90 | 9.57 | 30.88 |
| | | | Row | 155.15 | 87.03 | 48.19 | 222.47 | 5135.37 | 327.77 |
| | | | Column | 14.10 | 11.20 | 8.06 | 17.90 | 9.57 | 30.88 |
| | PTB | Centralized | - | 108.37 | 47.17 | 29.22 | 4567.49 | 115.68 | 240.14 |
| | | Local-only | - | 108.37 | 47.17 | 29.22 | 4567.49 | 115.68 | 240.14 |
| | | FedPrLLM | Layer | 108.37 | 47.17 | 29.22 | 4567.49 | 115.68 | 240.14 |
| | | | Row | 1060.91 | 394.57 | 239.91 | 21323.02 | 1075.71 | 928.73 |
| | | | Column | 108.37 | 47.17 | 29.22 | 4567.49 | 115.68 | 240.14 |

## B.3 MORE RESULTS ON OTHER LOCAL PRUNING METHODS

In this section, we provide additional experimental results using SparseGPT (Frantar & Alistarh, 2023), OWL (Yin et al., 2024), and BESA (Xu et al., 2024a) as the local pruning method to further validate the generality of our findings. For SparseGPT, we utilize the pruning metric proposed in SparseGPT (Frantar & Alistarh, 2023) and do not perform the weight update procedure (also adopted in Wanda (Sun et al., 2024); see Table 7 in (Sun et al., 2024)).

The experimental results of using other local pruning methods are shown in Tables 10 and 11. These results show a trend similar to those obtained using Wanda as the local pruning method and further demonstrate the generality of our findings.

## B.4 MORE RESULTS ON ZERO-SHOT TASKS

The experimental results on seven zero-shot tasks are shown in Tables 12 and 13. These results show a trend similar to those on the language modeling tasks and further demonstrate the generality of our findings.

Table 10: Perplexity (WikiText2 / C4 / PTB) of pruned LLaMA-7B under 50% sparsity ratio using other local pruning methods.

| Method | Compar. Group | Prune Stra. | Weight Scaling | SparseGPT (Frantar & Alistarh, 2023) | OWL (Yin et al., 2024) | BESA (Xu et al., 2024a) |
|---|---|---|---|---|---|---|
| Centralized | - | - | - | 7.40 / 9.54 / 76.18 | 7.21 / 9.31 / 67.44 | 7.27 / 9.34 / 78.74 |
| Local-only | - | - | - | 8.11 / 10.44 / 95.12 | 7.43 / 9.55 / 70.11 | 7.44 / 9.60 / 86.19 |
| FedPrLLM | Layer | One-shot | ✗ | **8.04** / **10.37** / 93.52 | 7.24 / **9.38** / **67.52** | 7.31 / **9.43** / **80.28** |
| | Row | One-shot | ✗ | 8.05 / **10.37** / **93.15** | **7.23** / 9.39 / 67.56 | 7.31 / **9.43** / 80.38 |
| | Column | One-shot | ✗ | 4279.74 / 4868.07 / 11451.43 | 1408.46 / 914.26 / 3338.93 | 1548.53 / 932.58 / 4683.50 |
| | Layer | Iterative | ✗ | **8.04** / **10.37** / 94.09 | **7.23** / 9.40 / 67.77 | **7.30** / **9.43** / 81.31 |
| | Row | Iterative | ✗ | 8.06 / **10.37** / **93.15** | **7.23** / 9.39 / 67.62 | **7.30** / 9.44 / 81.88 |
| | Column | Iterative | ✗ | 2562.72 / 4263.29 / 5643.11 | 1171.66 / 905.47 / 2100.39 | 1823.51 / 983.13 / 4909.44 |
| | Layer | One-shot | ✓ | 8.17 / 10.52 / 97.55 | 7.65 / 9.87 / 86.91 | 7.47 / 9.63 / 87.92 |
| | Row | One-shot | ✓ | 8.18 / 10.53 / 97.32 | 7.64 / 9.87 / 86.30 | 7.47 / 9.64 / 88.18 |
| | Column | One-shot | ✓ | 6524.84 / 7887.48 / 9790.79 | 1433.32 / 994.49 / 3598.38 | 1693.96 / 891.77 / 4662.56 |
| | Layer | Iterative | ✓ | 8.16 / 10.51 / 97.72 | 7.41 / 9.57 / 71.36 | 7.46 / 9.64 / 87.44 |
| | Row | Iterative | ✓ | 8.17 / 10.52 / 97.14 | 7.42 / 9.57 / 71.24 | 7.46 / 9.64 / 87.64 |
| | Column | Iterative | ✓ | 2741.71 / 3998.72 / 6088.04 | 1455.31 / 939.69 / 2790.60 | 2178.33 / 1147.38 / 8064.72 |

Table 11: Perplexity (WikiText2 / C4 / PTB) of pruned LLaMA-7B under 50% sparsity ratio when changing the comparison group for the local pruning method. FedPrLLM adopts one-shot pruning and no weight scaling.

| Local Compar. Group | Method | Compar. Group | SparseGPT (Frantar & Alistarh, 2023) | OWL (Yin et al., 2024) | BESA (Xu et al., 2024a) |
|---|---|---|---|---|---|
| Layer | Centralized | - | 7.91 / 10.21 / 83.25 | 7.61 / 9.88 / 71.59 | 7.94 / 10.28 / 92.81 |
| | Local-only | - | 8.89 / 11.58 / 108.47 | 7.84 / 10.12 / 76.15 | 8.16 / 10.56 / 99.17 |
| | FedPrLLM | Layer | **8.83** / **11.50** / **106.83** | **7.80** / **10.12** / **71.76** | **7.98** / **10.34** / **92.26** |
| | | Row | 183.63 / 134.18 / 913.14 | 10.54 / 13.40 / 124.99 | 32.54 / 35.30 / 291.07 |
| | | Column | 4623.88 / 4722.64 / 12115.99 | 1115.07 / 780.56 / 2480.77 | 1767.87 / 966.04 / 3964.13 |
| Column | Centralized | - | 8.86 / 14.10 / 108.37 | 7.89 / 10.82 / 72.35 | 8.23 / 11.64 / 100.07 |
| | Local-only | - | 8.86 / 14.10 / 108.37 | 7.91 / 10.86 / 73.27 | 8.89 / 14.19 / 109.73 |
| | FedPrLLM | Layer | **8.86** / **14.10** / **108.37** | **7.91** / 10.84 / **73.02** | **8.86** / 14.12 / **108.12** |
| | | Row | 138.54 / 155.15 / 1060.91 | 32.24 / 46.92 / 645.47 | 138.87 / 154.99 / 1064.28 |
| | | Column | **8.86** / **14.10** / **108.37** | **7.91** / **10.83** / **73.02** | **8.86** / **14.10** / 108.14 |

Table 12: Accuracies (%) on seven zero-shot tasks of pruned LLaMA-7B model under 50% sparsity ratio using Wanda as the local pruning method.

| Method | Compar. Group | Prune Stra. | Weight Scaling | HellaSwag | WinoGrande | OBQA | RTE | BoolQ | ARC-c | ARC-e | Mean |
|---|---|---|---|---|---|---|---|---|---|---|---|
| Dense | - | - | - | 56.96 | 70.09 | 34.20 | 66.43 | 75.11 | 41.89 | 75.29 | 59.99 |
| Centralized | - | - | - | 51.89 | 66.54 | 28.60 | 55.60 | 71.16 | 36.86 | 69.44 | 54.30 |
| Local-only | - | - | - | 51.52 | 66.23 | 28.55 | 55.37 | 70.85 | 36.49 | 69.13 | 54.02 |
| FedPrLLM | Layer | One-shot | ✗ | **51.93** | **66.61** | 29.80 | 53.49 | **71.22** | **37.03** | 69.49 | **54.22** |
| | Row | One-shot | ✗ | 51.84 | **66.61** | 30.20 | 53.07 | 71.16 | 36.77 | **69.61** | 54.18 |
| | Column | One-shot | ✗ | 26.24 | 50.51 | 13.60 | 52.35 | 38.01 | 20.65 | 30.56 | 33.13 |
| | Layer | Iterative | ✗ | **51.93** | 66.46 | 29.20 | 54.15 | 71.13 | 36.95 | **69.61** | 54.20 |
| | Row | Iterative | ✗ | 51.90 | 66.54 | 29.40 | 54.33 | 71.13 | 36.69 | 69.44 | 54.20 |
| | Column | Iterative | ✗ | 26.28 | 49.96 | 11.60 | 52.35 | 40.55 | 21.25 | 31.44 | 33.35 |
| | Layer | One-shot | ✓ | 51.42 | 66.51 | 30.20 | 53.07 | 71.19 | 36.60 | 68.98 | 54.00 |
| | Row | One-shot | ✓ | 51.80 | 66.33 | 30.20 | 53.79 | 71.10 | 36.30 | 69.16 | 54.09 |
| | Column | One-shot | ✓ | 25.92 | 50.12 | 12.00 | 51.26 | 38.62 | 20.14 | 29.46 | 32.50 |
| | Layer | Iterative | ✓ | 51.90 | 66.14 | 28.80 | 53.79 | 71.07 | 36.77 | 69.40 | 53.98 |
| | Row | Iterative | ✓ | 51.89 | 66.54 | 29.60 | 54.11 | 71.15 | 36.20 | 69.10 | 54.08 |
| | Column | Iterative | ✓ | 26.20 | 49.57 | 11.80 | 53.43 | 38.81 | 21.42 | 31.86 | 33.30 |

## B.5 RESULTS ON ULTRA-LOW CALIBRATION DATA REGIME

To further explore the performance of our FedPrLLM framework in scenarios with extremely limited calibration data (e.g., 1 sample/client), we conduct additional experiments using only 1 sample per client for calibration. We ran this challenging experiment on LLaMA-7B and LLaMA-2-7B with 128 clients (each holding only a single calibration sample) at 50% sparsity. For FedPrLLM, we use our recommended configuration of layer comparison, one-shot pruning, and no weight scaling. The results are presented in Table 14.

Table 13: Accuracies (%) on seven zero-shot tasks of pruned LLaMA-7B model under 50% sparsity ratio when changing the comparison group for the local pruning method (i.e., Wanda). FedPrLLM adopts one-shot pruning and no weight scaling.

| Local Compar. Group | Method | Compar. Group | HellaSwag | WinoGrande | OBQA | RTE | BoolQ | ARC-c | ARC-e | Mean |
|---|---|---|---|---|---|---|---|---|---|---|
| Layer | Centralized | - | 50.00 | 66.85 | 28.40 | 50.18 | 69.69 | 36.60 | 67.13 | 52.69 |
| | Local-only | - | 49.59 | 66.33 | 27.57 | 50.07 | 68.58 | 35.56 | 67.11 | 52.12 |
| | FedPrLLM | Layer | **50.04** | **65.59** | **27.80** | 49.82 | **68.72** | **36.09** | **67.93** | **52.28** |
| | | Row | 44.34 | 64.01 | 26.40 | 51.62 | 56.51 | 30.55 | 66.08 | 48.50 |
| | | Column | 25.78 | 50.91 | 12.20 | **52.35** | 37.95 | 20.82 | 27.86 | 32.55 |
| Column | Centralized | - | 48.92 | 65.82 | 26.20 | 56.68 | 65.11 | 34.56 | 66.79 | 52.01 |
| | Local-only | - | 48.92 | 65.82 | 26.20 | 56.68 | 65.11 | 34.56 | 66.79 | 52.01 |
| | FedPrLLM | Layer | **48.92** | **65.82** | **26.20** | **56.68** | **65.11** | **34.56** | **66.79** | **52.01** |
| | | Row | 35.60 | 56.20 | 20.80 | 53.43 | 50.95 | 26.28 | 60.23 | 43.35 |
| | | Column | **48.92** | **65.82** | **26.20** | **56.68** | **65.11** | **34.56** | **66.79** | **52.01** |

Table 14: Perplexity (WikiText2 / C4 / PTB) of pruned LLMs under 50% sparsity ratio in the ultra-low data regime (1 sample per client).

| Method | LLaMA-7B | LLaMA-2-7B |
|---|---|---|
| Centralized | 7.25 / 9.34 / 80.12 | 6.46 / 8.94 / 96.99 |
| Local-only | 7.58 / 9.73 / 89.22 | 6.77 / 9.30 / 116.60 |
| FedPrLLM | 7.31 / 9.46 / 82.33 | 6.49 / 9.04 / 103.24 |

As shown in Table 14, FedPrLLM consistently outperforms the Local-only baseline even in this ultra-low data regime. These results highlight the core strength of FedPrLLM: it effectively aggregates 128 individual masks into a single robust global mask, thereby overcoming the instability that severely impacts the Local-only approach.

## C   PRIVACY AND LEAKAGE ANALYSIS

To formally and empirically assess the privacy leakage of our framework, we conduct a detailed privacy analysis on the LLaMA-7B model, covering both theoretical limits and practical attack simulations.

To measure the maximum possible information leakage, we first perform an information entropy analysis. This tells us the theoretical limit of how much data a message can hold. Our analysis shows that a binary mask (at 50% sparsity) holds only 1.0 bit of information, while standard Float16 model weights hold 13.75 bits [3]. This means the mask contains only 7.3% of the information found in the weights—a 92.7% reduction. This massive reduction acts as a primary defense, making attack much harder because there is simply very little information available to leak.

Building on this theory, we test our mask-sharing method with a series of practical experiments. First, we check mask similarity to see if a mask is uniquely tied to the private data used to create it. We find that masks generated with Wanda using completely different, randomly seeded calibration data are over 95% identical (4.96% Hamming distance). This high similarity proves that the mask matrices are mostly determined by the public pre-trained model's weight, not the private data. This effectively separates the shared information from the private data. Next, our Differential Privacy (DP) (Dwork, 2006) sensitivity analysis shows that changing just one sample in the dataset causes a very small change in the mask matrices ($\sim$4.96% Hamming distance). Specifically, we create two datasets that differ by only one sample and measure the difference (i.e., Hamming distance) between their masks. This extremely low sensitivity means our method naturally provides strong privacy protection (equivalent to a formal privacy budget of $\epsilon \approx 0.05$) without needing to add extra noise.

---

[3] As calculating entropy across all model parameters is computationally prohibitive, this analysis compares data from a single sub-layer (`q_proj`) within the first transformer block.

We also simulate targeted attacks to test for privacy leakage. To test for Membership Inference Attacks (MIA) (Shokri et al., 2017), where an attacker tries to guess if a specific data record was used, we simulate a metric-based attack scenario. Since standard MIA relies on confidence scores (which our binary masks don't have), we measure the "signal strength"—the specific influence of a target sample on the final mask. We find that the difference in masks generated with and without a specific target sample is only 3.23% Hamming distance. This variation is smaller than the natural differences caused by using different datasets ($\sim$4.96%), making it difficult for an attacker to tell the difference between a real signal and random noise. This implies that any complex attack models would likely fail because the signal is too weak (Shokri et al., 2017; Dwork et al., 2006). Finally, we simulate a Gradient Inversion Attacks (Zhu et al., 2019; Fredrikson et al., 2015), where an attacker (e.g., an honest-but-curious server) with full knowledge of the model tries to reconstruct the original training data via gradient-based optimization. The attacker starts with a random noise tensor as input data and iteratively optimizes it to generate a mask that matches the target mask shared by the client. The loss function is the Hamming distance between the generated mask and the target mask. The gradients of this loss with respect to the input data are used to update the input, effectively "searching" for data that could produce the target mask. This attack also fails, recovering less than 2% of the tokens and producing meaningless text. For example, Original Text: "*your Apple AirPods and EarPods. Easy & hassle free installation. Earbuddyz must be removed to charge AirPods...*". Reconstructed Text: "*¡deuxTvekirection Readlarzug hecho pertelled h threat todos installah={blearsefw stories lookup...*". The attack fails because it tries to reverse a highly underdetermined, multi-stage information loss chain:

$$\text{Data} \rightarrow \textbf{Activations} \rightarrow \textbf{Scaler} \rightarrow \textbf{Importance Score} \rightarrow \textbf{Mask}.$$

Most steps in this chain is practically irreversible:

- **Activations $\rightarrow$ Scaler**: Activations across thousands of tokens are compressed into a single L2-norm statistic per neuron, losing all temporal and distributional information.
- **Importance Score $\rightarrow$ Mask**: The continuous, high-entropy importance scores are binarized via a threshold. All information about the magnitude of the scores is permanently destroyed; only a single bit (above or below threshold) remains.

An attacker trying to reverse this process faces a problem with an astronomical number of possible solutions. Given only the final 1-bit mask, it is computationally infeasible to reconstruct the specific data that initiated the chain. This confirms the security of our approach against even the most powerful adversaries.

Therefore, by sharing only low-information binary masks, our framework fundamentally reduces privacy risks and offers strong, practical privacy protection.

## D PRACTICAL EFFICIENCY, COMMUNICATION COST, AND RESOURCE USAGE

This section complements our main results with a thorough analysis of computation time, communication costs across diverse network conditions, client heterogeneity, memory usage, and energy implications. Unless stated otherwise, all simulations are conducted using LLaMA-7B.

### D.1 PRUNING RUNTIME AND PEAK MEMORY

Table 15 reports the pruning runtime and peak memory across all evaluated methods. One-shot and iterative variants exhibit similar local pruning time on GPU (approximately 145 seconds), as both compute Hessians and sort importance scores. The primary difference between these strategies lies in the number of communication rounds: one-shot requires a single round (uploading masks once), whereas iterative requires one round per layer (32 rounds for LLaMA-7B).

Regarding memory usage, one-shot pruning shows higher peak memory (about 31 GB) than iterative (about 19 GB) in our single-machine simulation because the server aggregates masks across all layers simultaneously. In a real distributed deployment, masks can be processed in a streaming,

Table 15: Runtime and Peak Memory usage for all evaluated methods.

| Method | Compar. Group | Prune Stra. | Weight Scaling | Pruning Time (s) | Peak Memory (GB) |
|---|---|---|---|---|---|
| Centralized | - | - | - | 79.8 | 18.66 |
| Local-only | - | - | - | 142.5 | 25.14 |
| FedPrLLM | Layer | One-shot | ✗ | 143.2 | 31.27 |
| | Row | One-shot | ✗ | 143.8 | 31.27 |
| | Column | One-shot | ✗ | 142.9 | 31.27 |
| | Layer | Iterative | ✗ | 145.6 | 19.04 |
| | Row | Iterative | ✗ | 144.8 | 19.04 |
| | Column | Iterative | ✗ | 143.4 | 19.02 |
| | Layer | One-shot | ✓ | 143.5 | 31.27 |
| | Row | One-shot | ✓ | 144.1 | 31.27 |
| | Column | One-shot | ✓ | 143.0 | 31.27 |
| | Layer | Iterative | ✓ | 144.8 | 19.69 |
| | Row | Iterative | ✓ | 144.8 | 19.04 |
| | Column | Iterative | ✓ | 144.8 | 19.69 |

layer-by-layer fashion on the server, distributing the memory load across clients and reducing peak memory to be comparable to the iterative approach.

## D.2 BANDWIDTH VS. LATENCY TRADE-OFFS

We simulate end-to-end pruning time under four representative network profiles to quantify the interplay between bandwidth and latency. Table 16 summarizes the results. We observe that one-shot pruning method consistently achieves a ∼31x speedup over iterative pruning across all network conditions. This significant reduction in communication rounds makes one-shot pruning particularly advantageous in high-latency, low-bandwidth environments, such as edge networks.

Table 16: Simulated total communication time under different network conditions.

| Network Profile | Latency | Bandwidth | One-shot Time (h) | Iterative Time (h) | Speedup |
|---|---|---|---|---|---|
| Datacenter | 1ms | 10 Gbps | ∼0.1 | ∼3.1 | ∼31x |
| Cross-Silo (LAN) | 5ms | 1 Gbps | ∼1.0 | ∼31.3 | ∼31x |
| Cross-Silo (WAN) | 50ms | 100 Mbps | ∼9.9 | ∼313.3 | ∼31x |
| Edge | 100ms | 10 Mbps | ∼99.4 | ∼3132.9 | ∼31x |

## D.3 SYSTEM HETEROGENEITY (STRAGGLERS)

We further simulate system heterogeneity with 20% stragglers (slow clients) to compare the communication time of One-shot and Iterative pruning. Specifically, we instantiate 64 clients, where 51 *"fast"* clients finish the mask upload in 534.1 seconds, while 13 *"slow"* clients (bandwidth at 50%) take 1,068.2 seconds. In this setting, One-shot pruning incurs a +534 second straggler penalty only once, resulting in a total straggler overhead of 534 seconds (100% of the homogeneous upload time). By contrast, the iterative baseline must absorb the same 534-second penalty at every communication round; with 32 rounds, this compounds to $32 \times 534 \approx 17,090$ additional seconds (>4.7 hours) of idle time. This dramatic gap makes One-shot inherently robust to the system heterogeneity typical of cross-device federated learning.

## D.4 COMPREHENSIVE EFFICIENCY, SCALABILITY, AND ENERGY

Tablere 17 summarizes time efficiency, scalability under heterogeneity, energy implications, memory/storage, and theoretical inference metrics for LLaMA-7B in a Cross-Silo WAN environment (100,Mbps bandwidth, 50,ms latency). One-shot pruning reduces total pruning time by, which dominates energy consumption in federated settings and translates to energy savings. In terms of scalability, one-shot suffers the straggler penalty only once, whereas iterative methods incur it in every

round, making one-shot substantially more robust. Both methods achieve equivalent storage compression at the same sparsity.

For inference time, unstructured sparsity reduces the theoretical FLOPs of pruned layers (e.g., at 50% sparsity), but practical speedups on standard GPUs may require specialized sparse kernels. Realizing hardware-level inference acceleration is complementary to and beyond the scope of this work.

Table 17: Comprehensive analysis of efficiency, scalability, and resource usage. Note: Energy savings (>90%) are derived from the 31x reduction in total communication time, which dominates the energy consumption in federated settings.

| Metric Category | Specific Metric | One-shot | Iterative | Improvement |
|---|---|---|---|---|
| Time Efficiency | Total Pruning Time | 9.9 hours | > 313 hours | 31x Speedup |
| | Straggler Impact | 1x Penalty (Once) | 32x Penalty (Every Layer) | Robust |
| Energy | Pruning Energy Cost | Low | Very High | > 90% Savings |
| Memory | Model Size (Storage) | 6.5 GB | 6.5 GB | Equivalent |
| Inference | Theoretical Throughput | 1.43x | 1.43x | Equivalent |
| | Theoretical Latency | 0.70x | 0.70x | Equivalent |

# E    THE USE OF LARGE LANGUAGE MODELS (LLMs)

We used Large Language Models (LLMs) to enhance the language and clarity of this manuscript. Their role included rephrasing for readability, correcting grammatical errors, and ensuring consistent terminology. All core scientific contributions, including the proposed methods, experimental design, and results analysis, are original to the authors. The LLMs acted solely as writing assistants and did not influence the research ideas or outcomes presented.

