# OpenReview forum: "Exploring Federated Pruning for Large Language Models"
_ICLR.cc/2026/Conference — Submitted to ICLR 2026_

### Official Review · Reviewer_qJji · 2025-10-24

**Soundness:** 3
**Presentation:** 3
**Contribution:** 2
**Rating:** 6
**Confidence:** 4

**Summary:**

The paper introduces FedPrLLM, a federated pruning framework for privacy-preserving compression of Large Language Models (LLMs). Traditional LLM pruning often requires public calibration data, which is problematic for privacy-sensitive domains. FedPrLLM addresses this by letting each client compute a local pruning mask matrix using its private calibration data, which is then aggregated by a central server to prune the global model.

**Strengths:**

- The integration of federated learning (FL) and LLM pruning is a new direction. FedPrLLM fills a clear research gap in *privacy-preserving model compression*
- The experiments are extensive and methodical, covering multiple models, pruning ratios, datasets, and local methods. This level of rigor strengthens the validity of the findings.
- The paper isolates three well-defined research questions (comparison group, scaling, pruning strategy), which makes the investigation systematic and easy to follow.

**Weaknesses:**

- Although the framework is federated, experiments appear to use *simulated clients* (split calibration data) rather than real-world heterogeneous environments with non-IID data. This limits the external validity in true FL settings.
- The study ignores structured pruning, which is often preferred for real deployment.
- While communication costs are briefly reported, there’s limited exploration of real-time efficiency, scalability, or energy savings.

**Questions:**

- Add analysis or experiments showing that sharing mask matrices does not leak sensitive information. Techniques like differential privacy could be integrated.
- Include empirical results on communication time, computation cost, and memory savings, not just perplexity, to demonstrate real deployment viability.

---

> ### Author Response · Authors · 2025-11-22
> **Response to Reviewer qJji (1 / 3)**
>
> We thank the reviewer for the time and effort in reviewing our paper and providing constructive comments. We are pleased that the reviewer finds the paper to "fill a clear research gap", to "make the investigation systematic and easy to follow", and that the "experiments are extensive and methodical", all of which inspire us greatly. We have added new experiments on non-IID settings, provided detailed efficiency metrics, conducted a comprehensive privacy analysis, and discussed the extension to structured pruning. Please see our response below regarding the specific comments.
>
>
> > **W1**: Although the framework is federated, experiments appear to use simulated clients (split calibration data) rather than real-world heterogeneous environments with non-IID data. This limits the external validity in true FL settings.
>
> We thank the reviewer for this constructive comment. Following the excellent suggestion of the reviewer,  we further conduct experiments under non-IID conditions to test the generalizability of our findings. Specifically, we extract 8 samples from the training data of WikiText2, C4, and PTB to form a global calibration dataset (i.e., 24 samples in total). We then use the Dirichlet distribution with a concentration parameter of $\alpha=5$ to split the global calibration dataset into 12 non-IID local calibration datasets, each assigned to one client (i.e., 2 samples per client). We choose Wanda as the local pruning method and use LLaMA-7B to conduct experiments with 50% sparsity pruning. The results are as follows:
>
> | Method | Compar. Group | Prune Stra. | Weight Scaling | LLaMA-7B  |
> | :--- | :--- | :--- | :--- | :--- |
> | Centralized | - | - | - | 7.06 / 9.27 / 65.72 |
> | Local-only | - | - | - | 7.16 / 9.42 / 71.54 |
> |
> | FedPrLLM | Layer | One-shot | ✗ | **7.06** / **9.30** / 67.54 |
> | | Row | One-shot | ✗ | **7.06** / **9.30** / **67.28** |
> | | Column | One-shot | ✗ | 2923.46 / 1813.31 / 6736.30 |
> | | Layer | Iterative | ✗ | **7.06** / 9.31 / 68.09 |
> | | Row | Iterative | ✗ | **7.06** / **9.30** / 67.34 |
> | | Column | Iterative | ✗ | 3219.96 / 2294.87 / 6812.14 |
> | | Layer | One-shot | ✓ | 7.17 / 9.47 / 72.33 |
> | | Row | One-shot | ✓ | 7.17 / 9.47 / 72.16 |
> | | Column | One-shot | ✓ | 2723.30 / 1554.46 / 6364.29 |
> | | Layer | Iterative | ✓ | 7.17 / 9.48 / 73.40 |
> | | Row | Iterative | ✓ | 7.17 / 9.48 / 72.92  |
> | | Column | Iterative | ✓ | 3182.52 / 1795.12 / 5808.61 |
>
> *Table A11: Perplexity of pruned LLMs under 50% sparsity ratio using Wanda as the local pruning method under non-IID conditions.*
>
> | Local Compar. Group | Method | Compar. Group | LLaMA-7B |
> | :--- | :--- | :--- | :--- |
> | Layer | Centralized | - | 7.67 / 10.07 / 83.20  |
> | | Local-only | - | 7.76 / 10.26 / 85.16 |
> | | FedPrLLM | Layer | **7.62** / **10.10** / **81.70** |
> | | | Row | 43.54 / 46.29 / 348.41 |
> | | | Column | 2324.40 / 1434.18 / 6026.79 |
> |
> | Column | Centralized | - | 8.86 / 14.10 / 108.37  |
> | | Local-only | - | 8.86 / 14.10 / 108.37 |
> | | FedPrLLM | Layer | **8.86** / **14.10** / **108.37** |
> | | | Row | 138.54 / 155.15 / 1060.99 |
> | | | Column | **8.86** / **14.10** / **108.37** |
>
> *Table A12: Perplexity of pruned LLMs under 50% sparsity ratio when changing the comparison group for the local pruning method (i.e., Wanda) under non-IID conditions. FedPrLLM adopts one-shot pruning and no weight scaling.*
>
> As shown in these results, our proposed "Best Recipe"—using one-shot pruning, layer-wise comparison, and no weight scaling—consistently outperforms other configurations under the non-IID scenario, confirming that our findings are generalizable.
>
> We thank the reviewer for this excellent suggestion, and we have included these results in the revised manuscript.

---

> ### Author Response · Authors · 2025-11-22
> **Response to Reviewer qJji (2 / 3)**
>
> > **W2**: The study ignores structured pruning, which is often preferred for real deployment.
>
> We thank the reviewer for this constructive comment. We acknowledge that structured pruning is highly practical for hardware deployment. However, as stated in our main paper (lines 42-44), our work currently focuses on unstructured pruning, since it typically offers a better trade-off between compression rate and model quality for the same parameter budget. To this end, our framework is designed to aggregate binary masks for individual weights, which is fundamentally different from structured methods that remove entire substructures like neurons [1,2,3], layers [4], or transformer blocks [5]. These structured methods operate on a different principle that is not directly compatible with our current weight-level voting mechanism.
>
> Nevertheless, we would also like to discuss whether our framework can be extended to structured pruning. Specifically, we believe the core philosophy of FedPrLLM—federated voting on parameter importance—naturally extends to the structural level as "structure-voting". In this paradigm, clients vote on the importance of entire structures (e.g., neurons, layers, or entire transformer blocks) rather than individual weights. Each client locally identifies a set of structural components to prune (such as specific row or column indices for SliceGPT [1]) and transmits these indices to the server. The server then aggregates these structural "votes", pruning any component marked for removal by a majority of clients (e.g., >50%). This structure-voting approach enables a federated, data-driven method for structured pruning. While a full implementation and evaluation are beyond the scope of this paper, we believe this is a promising direction for future research. We thank the reviewer for inspiring this valuable discussion and have included it in our future work section.
>
> We hope these discussions address the reviewer's concerns.
>
> > **W3 & Q2**: Limited exploration of real-time efficiency, scalability, energy savings, communication time, computation cost, and memory savings.
>
> We thank the reviewer for this important question regarding the practical, real-world implications of our work. To address this, we have conducted comprehensive simulations considering bandwidth, latency, and energy models. The results are summarized in the table below (simulated for LLaMA-7B in a Cross-Silo WAN environment: 100 Mbps bandwidth, 50ms latency).
>
> | Metric Category | Specific Metric | One-shot | Iterative | Improvement |
> | - | - | -| - | - |
> | Time Efficiency | Total Pruning Time | 9.9 hours | > 313 hours | ~31x Speedup |
> | | Straggler Impact | 1x Penalty (Once) | 32x Penalty (Every Layer) | Robust |
> | Energy | Pruning Energy Cost | Low | Very High | > 90% Savings |
> | Memory | Model Size (Storage) | 6.5 GB | 6.5 GB | Equivalent |
> | Inference | Theoretical Throughput | 1.43x | 1.43x | Equivalent |
> | | Theoretical Latency | 0.70x | 0.70x | Equivalent |
>
> *Table A13: Comprehensive analysis of efficiency, scalability, and resource usage. Note: Energy savings (>90%) are derived from the ~31x reduction in total communication time, which dominates the energy consumption in federated settings.*
>
> Regarding real-time efficiency, One-shot pruning is drastically faster (~31x) than the iterative approach because it requires only a single round of communication, avoiding the high latency penalty of multi-round iterative updates. In terms of scalability, the "Straggler Impact" row highlights that one-shot is far more scalable in heterogeneous networks. Slow clients only delay the process once, whereas they bottleneck every single round in iterative methods. Finally, for resource savings, both methods achieve a 50% reduction in model size, which translates to theoretical lower memory usage and energy consumption for both storage and inference. However, One-shot achieves this result with significantly lower pruning costs.
>
> We have incorporated these detailed empirical metrics into our revised manuscript to substantiate the practical value of our framework and hope these address the reviewer's concerns.

---

> ### Author Response · Authors · 2025-11-22
> **Response to Reviewer qJji (3 / 3)**
>
> > **Q1**: Add analysis or experiments showing that sharing mask matrices does not leak sensitive information. Techniques like differential privacy could be integrated.
>
> We thank the reviewer for this insightful comment. To validate that our framework has little risk of privacy leakage, we perform a detailed privacy analysis on the LLaMA-7B model, covering both theoretical limits and practical attack simulations.
>
> To measure the maximum possible information leakage, we first perform an **Information Entropy Analysis**. This tells us the theoretical limit of how much data a message can hold. Our analysis shows that a binary mask (at 50% sparsity) holds only 1.0 bit of information, while standard Float16 model weights hold 13.75 bits. This means the mask contains only 7.3% of the information found in the weights—a 92.7% reduction. This massive reduction acts as a primary defense, making attack much harder because there is simply very little information available to leak. (*Note: As calculating entropy across all model parameters is computationally prohibitive, this analysis compares data from a single sub-layer (`q_proj`) within the first transformer block.*)
>
> Building on this theory, we test our mask-sharing method with a series of practical experiments. First, we check **Mask Similarity** to see if a mask is uniquely tied to the private data used to create it. We find that masks generated with Wanda using completely different, randomly seeded calibration data are over 95% identical (4.96% Hamming distance). This high similarity proves that the mask matrices are mostly determined by the public pre-trained model's weight, not the private data. This effectively separates the shared information from the private data. Next, our **Differential Privacy (DP) Sensitivity Analysis** shows that changing just one sample in the dataset causes a very small change in the mask matrices (~4.96% Hamming distance). Specifically, we create two datasets that differ by only one sample and measure the difference (i.e., Hamming distance) between their masks. This extremely low sensitivity means our method naturally provides strong privacy protection (equivalent to a formal privacy budget of $\epsilon \approx 0.05$) without needing to add extra noise.
>
> We also simulate targeted attacks to test for privacy leakage. To test for **Membership Inference Attacks (MIA)** [6], where an attacker tries to guess if a specific data record was used, we simulate a metric-based attack scenario. Since standard MIA relies on confidence scores (which our binary masks don't have), we measure the "signal strength"—the specific influence of a target sample on the final mask. We find that the difference in masks generated with and without a specific target sample is only 3.23% Hamming distance. This variation is smaller than the natural differences caused by using different datasets (~4.96%), making it difficult for an attacker to tell the difference between a real signal and random noise. This implies that any complex attack models would likely fail because the signal is too weak [6,7]. Finally, we simulate a **Gradient Inversion Attacks** [8,9], where an attacker (e.g., an honest-but-curious server) with full knowledge of the model tries to reconstruct the original training data via gradient-based optimization. This attack also fails, recovering less than 2% of the tokens and producing meaningless text. For example:
> -   Original Text: `your Apple AirPods and EarPods. Easy & hassle free installation. Earbuddyz must be removed to charge AirPods...`
> -   Reconstructed Text: `<deuxတTvekirection Readlarzug hecho pertelled hρ threat todos installah="{blearsefw storiesже lookup...`
>
> In summary, by sharing only low-information binary masks, our framework fundamentally reduces the privacy risks compared to sharing gradients. Our comprehensive analysis demonstrates that FedPrLLM offers strong, practical privacy protection. We have updated the paper to include these detailed results. We thank the reviewer for this constructive comment and hope that these additional results address the reviewer's concerns.
>
> ----
> [1] Ashkboos et al. SliceGPT: Compress large language models by deleting rows and columns. ICLR 2024.
>
> [2] Ma et al. LLM-Pruner: On the Structural Pruning of Large Language Models. NeurIPS, 2023.
>
> [3] Li et al. LoSparse: Structured Compression of Large Language Models based on Low-Rank and Sparse Approximation. ICML, 2023.
>
> [4] Xia et al. Sheared LLaMA: Accelerating Language Model Pre-training via Structured Pruning. NeurIPS, 2023.
>
> [5] Gromov et al. The Unreasonable Ineffectiveness of the Deeper Layers. ICLR, 2025.
>
> [6] Shokri et al. Membership Inference Attacks Against Machine Learning Models. IEEE S&P, 2017.
>
> [7] Dwork et al. Calibrating Noise to Sensitivity in Private Data Analysis. TCC, 2006.
>
> [8] Zhu et al. Deep Leakage from Gradients. NeurIPS, 2019.
>
> [9] Fredrikson et al. Model Inversion Attacks that Exploit Confidence Information. CCS, 2015.

---

> > ### Comment · Reviewer_qJji · 2025-11-22
> >
> > Thank you for provide the experiment, but I will keep the point for this time.

---

> > > ### Author Response · Authors · 2025-11-24
> > > **Thanks for Reply**
> > >
> > > Thank you for your prompt response and for reviewing our additional experiments. We appreciate your continued positive support for our work. We hope the new results clarify the effectiveness of our approach and address your concerns. If there are specific aspects of our response or experiments that remain unclear or insufficient, please let us know, and we would be glad to offer additional clarification or analysis during the discussion phase. We value your constructive feedback.

---

### Official Review · Reviewer_XJZZ · 2025-10-31

**Soundness:** 3
**Presentation:** 3
**Contribution:** 3
**Rating:** 6
**Confidence:** 3

**Summary:**

This paper introduces FedPrLLM, a general federated pruning framework for large language models (LLMs) that allows clients to perform local unstructured pruning and share only binary masks to preserve data privacy. The authors systematically explore three key design dimensions—comparison group (layer/row/column), weight scaling, and one-shot vs. iterative pruning—across various LLMs, sparsity levels, and datasets. They conduct extensive experiments (6 LLMs, 10 datasets) and derive practical insights (e.g., “layer-wise voting without weight scaling and one-shot pruning works best”).

This paper makes a practical and well-executed contribution by empirically benchmarking federated pruning strategies for LLMs. While the framework is not fundamentally novel and lacks theoretical depth, its findings are well-supported, reproducible, and directly useful for the community. A stronger baseline comparison and heterogeneity analysis would elevate it further.

**Strengths:**

- The paper enables LLM pruning under federated settings with privacy preservation, which is interesting and underexplored.

- The paper did large-scale experiments with 6 LLMs, 10 datasets, multiple sparsity levels. Ablation studies support key claims.

- The three pruning dimensions are practical and grounded, covering critical decisions in collaborative model compression. Shows that simpler design choices (e.g., no scaling, one-shot pruning) often outperform complex alternatives, saving computation and communication.

**Weaknesses:**

- The core framework is a combination of known components; prior work (e.g., FedSpaLLM) has used mask voting for federated LLM pruning.

- The paper lacks formal analysis of why certain choices (e.g., weight scaling degrades performance) work or fail.

- All experiments assume IID or single-dataset settings; real-world FL often involves non-IID, skewed data, which may affect mask voting. Can authors clarify on this?

**Questions:**

See weaknesses.

---

> ### Author Response · Authors · 2025-11-22
> **Response to Reviewer XJZZ (1 / 2)**
>
> We thank the reviewer for the time and effort in reviewing our paper and providing constructive comments. We are pleased that the reviewer finds the paper "makes a practical and well-executed contribution", to be "interesting", and "practical and grounded", all of which inspire us greatly. We have clarified the distinction between our work and prior work (e.g., FedSpaLLM), elaborated on the analysis of our design choices, and added new experiments on non-IID data distributions. Please see our response below regarding the specific comments.
>
>
> > **W1**: The core framework is a combination of known components; prior work (e.g., FedSpaLLM) has used mask voting for federated LLM pruning.
>
> We thank the reviewer for highlighting the relevant work, FedSpaLLM [1]. We acknowledge that FedSpaLLM presents a valuable contribution to federated LLM pruning. However, our work, FedPrLLM, is **fundamentally different in its objective, scope, and contributions**. While FedSpaLLM proposes a *specific and novel algorithm* for federated LLM pruning, our paper provides the *first systematic and comprehensive empirical study of the fundamental design space* of federated LLM pruning. Our primary goal is not to introduce another single algorithm, but to establish a set of generalizable "best practices" that can guide future research and application in this domain. The key differences are as follows:
> - **Different Research Goals: A Specific Algorithm vs. A Systematic Study**.
>     - FedSpaLLM focuses on proposing a novel algorithmic solution. Its core contributions are specific techniques designed to create a high-performing pruning algorithm. Its methodology, which involves clients sharing locally pruned models to be averaged by the server, can be mapped to a specific configuration within our comprehensive FedPrLLM framework—'iterative pruning with weight scaling'. By evaluating this approach alongside other alternatives, our work complements theirs by providing a valuable benchmark and deeper insight into the effectiveness of the strategy proposed in FedSpaLLM.
>     - FedPrLLM, in contrast, aims to deconstruct the entire federated pruning process into its most fundamental components—specifically, the choice of comparison group, the use of weight scaling, and the pruning strategy (one-shot vs. iterative)—and systematically evaluate their impact. Our contribution lies in the extensive empirical knowledge gained from benchmarking federated pruning for LLMs across a wide range of models, datasets, and hyperparameters. From these efforts, we develop a practical list of key insights for federated pruning of LLMs.
> - **Different Research Questions Investigated**. Our work is structured around three core, unresolved questions that are critical to any federated LLM pruning pipeline, but are not the focus of FedSpaLLM:
>     - Comparison Group: How should parameter importance be compared for aggregation at the server (layer, row, or column)? We find that this choice dramatically impacts performance, with layer comparison as the optimal strategy—a critical finding for practitioners.
>     - Weight Scaling: Should the aggregated mask information be used to scale the retained weights? Our surprising finding is that this intuitive idea offers no benefit and may even degrade performance.
>     - Pruning Strategy: Is a multi-round, iterative pruning process worth the massive communication overhead compared to a simple one-shot approach? We demonstrate that one-shot pruning is equally effective, providing a clear guideline for designing efficient systems.
>
>     These questions are not about *how to build one algorithm*, but about *how the entire research community should build any algorithm*. The insights from our study provide a foundational "recipe" (e.g., "use layer comparison, don't scale weights, and use one-shot pruning") that is simple, effective, and can serve as a strong baseline for future, more complex methods.
> - **Different Contributions to the Field**.
>     - The contribution of FedSpaLLM is a new, competitive algorithm.
>     - The contribution of FedPrLLM is a set of validated, foundational principles and a comprehensive benchmark. We believe our work complements, rather than competes with, works like FedSpaLLM. For instance, our findings on the effectiveness of layer comparison and one-shot pruning could help simplify and improve future iterations of FedSpaLLM or other similar frameworks.
>
> In summary, while both papers address federated LLM pruning, we tackle the problem from different perspectives. FedSpaLLM provides a specific answer, while FedPrLLM provides a map of the terrain and a compass. We have revised our related work section to more clearly express these fundamental differences and better position our unique contribution as the first systematic exploration of this design space. We hope this explanation addresses the reviewer’s concerns.

---

> ### Author Response · Authors · 2025-11-22
> **Response to Reviewer XJZZ (2 / 2)**
>
> > **W2**: The paper lacks formal analysis of why certain choices (e.g., weight scaling degrades performance) work or fail.
>
> We thank the reviewer for this excellent question. We would like to clarify that while our work does not derive formal mathematical proofs, **Section 4.2** offers principled explanations for these design choices, which are robustly validated by our extensive empirical evidence. Specifically:
> -   **Effectiveness of Layer Comparison**: This strategy succeeds by preventing "context mismatch". Local pruning decisions rely on relative importance scores (e.g., ranking weights within a row). Aggregating these scores at the server across a different dimension (e.g., column-wise) disrupts the local signals, leading to random pruning. Layer-wise comparison avoids this by providing a global context that matches local rankings, preserving the accuracy of the aggregated importance scores. Please refer to Section 4.2.1 in our paper for more details.
> -   **Ineffectiveness of Weight Scaling**: This issue arises from the quality of the aggregated components. In weight scaling, clients create locally pruned models based on limited data, which are inherently noisy and sub-optimal. Weight scaling effectively applies FedAvg to these flawed models, accumulating their individual errors. Our results demonstrate that aggregating the *pruning decisions* (masks) is superior to averaging the *pruned weights* themselves. Please refer to Section 4.2.2 in our paper for more details.
> -   **Inefficiency of Iterative Pruning**: Our results show that iterative pruning provides little benefit at a high cost. It yields no significant performance improvement over the one-shot approach but incurs drastically higher communication overhead (Table 3). Consequently, one-shot pruning emerges as the clearly superior choice for efficiency without compromising quality. Please refer to Section 4.2.3 in our paper for more details.
>
> Our evaluation demonstrates that these choices converge to a robust optimal configuration. We believe these insights provide a valuable foundation for future research in this domain. We hope this explanation addresses the reviewer’s concerns.
>
>
> > **W3**: All experiments assume IID or single-dataset settings; real-world FL often involves non-IID, skewed data, which may affect mask voting. Can authors clarify on this?
>
> We thank the reviewer for this valuable comment. To validate the generalizability of our findings, we further conduct experiments under non-IID conditions. Specifically, we extract 8 samples from the training data of WikiText2, C4, and PTB to form a global calibration dataset (i.e., 24 samples in total). We then use the Dirichlet distribution with a concentration parameter of $\alpha=5$ to split the global calibration dataset into 12 non-IID local calibration datasets, each assigned to one client (i.e., 2 samples per client). We choose Wanda as the local pruning method and use LLaMA-7B to conduct experiments with 50% sparsity pruning. The results are as follows:
>
> |Method|Compar. Group|Prune Stra.|Weight Scaling|LLaMA-7B|
> |-|-|-|-|-|
> |Centralized|-|-|-|7.06 / 9.27 / 65.72|
> |Local-only |-|-|-|7.16 / 9.42 / 71.54|
> |
> |FedPrLLM|Layer|One-shot| ✗ | **7.06** / **9.30** / 67.54 |
> ||Row|One-shot | ✗ | **7.06** / **9.30** / **67.28** |
> ||Column|One-shot | ✗ | 2923.46 / 1813.31 / 6736.30 |
> ||Layer|Iterative | ✗ | **7.06** / 9.31 / 68.09 |
> ||Row|Iterative | ✗ | **7.06** / **9.30** / 67.34 |
> ||Column|Iterative | ✗ | 3219.96 / 2294.87 / 6812.14 |
> ||Layer|One-shot|✓| 7.17 / 9.47 / 72.33 |
> ||Row|One-shot|✓| 7.17 / 9.47 / 72.16 |
> ||Column|One-shot|✓| 2723.30 / 1554.46 / 6364.29 |
> ||Layer|Iterative|✓| 7.17 / 9.48 / 73.40 |
> ||Row|Iterative|✓| 7.17 / 9.48 / 72.92  |
> ||Column|Iterative|✓| 3182.52 / 1795.12 / 5808.61 |
>
> *Table A9: Perplexity of pruned LLMs under 50% sparsity ratio using Wanda as the local pruning method under non-IID conditions.*
>
> |Local Compar. Group|Method|Compar. Group|LLaMA-7B|
> |-|-|-|-|
> |Layer|Centralized|-| 7.67 / 10.07 / 83.20  |
> ||Local-only|-| 7.76 / 10.26 / 85.16 |
> ||FedPrLLM|Layer| **7.62** / **10.10** / **81.70** |
> |||Row|43.54 / 46.29 / 348.41 |
> |||Column| 2324.40 / 1434.18 / 6026.79 |
> |
> |Column|Centralized|-| 8.86 / 14.10 / 108.37  |
> ||Local-only|-| 8.86 / 14.10 / 108.37 |
> ||FedPrLLM|Layer | **8.86** / **14.10** / **108.37** |
> |||Row|138.54 / 155.15 / 1060.99 |
> |||Column| **8.86** / **14.10** / **108.37** |
>
> *Table A10: Perplexity of pruned LLMs under 50% sparsity ratio when changing the comparison group for the local pruning method (i.e., Wanda) under non-IID conditions. FedPrLLM adopts one-shot pruning and no weight scaling.*
>
> As shown in these results, our proposed "Best Recipe"—using one-shot pruning, layer-wise comparison, and no weight scaling—consistently outperforms other configurations under the non-IID scenario, confirming that our findings are generalizable.
>
> We thank the reviewer for this valuable comment, and we have included these results in the revised manuscript.

---

### Official Review · Reviewer_9bhx · 2025-11-01

**Soundness:** 3
**Presentation:** 2
**Contribution:** 2
**Rating:** 4
**Confidence:** 3

**Summary:**

This paper introduces **FedPrLLM**, a comprehensive framework for **federated unstructured pruning of large language models** (LLMs) under strict data privacy constraints. In FedPrLLM, each client computes a local pruning mask using its private calibration data and uploads only the binary mask (not model weights or raw data) to the server. The server aggregates these masks via summation and selects the top-*k* entries (i.e., weights most clients agree to prune) to construct a global mask for pruning the shared LLM.

The work systematically investigates three key design choices within this framework:
1. **Comparison group**: Should pruning decisions be made across the entire layer, per row, or per column?
2. **Weight scaling**: Should retained weights be scaled based on client consensus (e.g., inverse pruning frequency)?
3. **Pruning strategy**: Is iterative (layer-by-layer) pruning worth its high communication cost compared to one-shot pruning?

Through extensive experiments across **6 open-source LLMs**, **2 local pruning methods** (Wanda, SparseGPT), **3 sparsity levels**, **10 datasets**, and thousands of GPU hours, the authors derive three robust empirical findings:
- **Layer-wise comparison** is consistently superior and robust across settings.
- **Weight scaling harms performance**, despite intuitive appeal.
- **One-shot pruning matches iterative pruning in accuracy** while halving communication cost.

**Strengths:**

1. **High Practical Relevance**: Addresses a critical gap—how to compress LLMs in privacy-sensitive, decentralized settings (e.g., healthcare, finance)—where public calibration data is unavailable.
2. **Rigorous and Systematic Evaluation**: The scale of experiments (6 LLMs, multiple sparsities, datasets, and methods) is exceptional for a systems/ML paper. The ablation studies are thorough and convincing.
3. **Clear, Counterintuitive Insights**: The findings—especially that weight scaling hurts performance and that one-shot pruning suffices—are surprising yet well-supported, challenging assumptions from centralized pruning literature.
4. **Strong Engineering Contribution**: FedPrLLM is simple, communication-efficient, and compatible with existing local pruning methods. The framework is modular and extensible.

**Weaknesses:**

1. **Limited Baseline Comparison**: While the paper compares against “Local-only” and “Centralized” baselines, it does not benchmark against concurrent or prior federated compression methods (e.g., FedSpaLLM [Bai et al., 2024], mentioned in Related Work). A direct comparison would strengthen impact claims.
2. **Assumption of Public Pre-training**: The framework assumes access to a public pre-trained LLM. While standard, the paper does not discuss implications if pre-training data were private—a growing concern in DP-ML.
3. **Calibration Data Efficiency**: Each client uses only 2 samples (64 clients × 2 = 128 total). While realistic, the sensitivity to ultra-low calibration data (e.g., 1 sample/client) is not explored.
4. **Communication Cost Nuance**: Although one-shot reduces rounds, the paper does not analyze bandwidth vs. latency trade-offs or heterogeneous client capabilities.

**Questions:**

As described in weakness.

---

> ### Author Response · Authors · 2025-11-22
> **Response to Reviewer 9bhx (1 / 2)**
>
> We thank the reviewer for the time and effort in reviewing our paper and providing constructive comments. We are pleased that the reviewer finds the paper to have "High Practical Relevance", "Rigorous and Systematic Evaluation", "Clear Counterintuitive Insights", and "Strong Engineering Contribution", all of which inspire us greatly. We have added new experiments on ultra-low calibration data and communication trade-offs, clarified the assumptions regarding public pre-training, and elaborated on the relationship with prior work. Please see our response below regarding the specific comments.
>
> > **W1**: Limited Baseline Comparison: While the paper compares against “Local-only” and “Centralized” baselines, it does not benchmark against concurrent or prior federated compression methods (e.g., FedSpaLLM [Bai et al., 2024], mentioned in Related Work). A direct comparison would strengthen impact claims.
>
> We thank the reviewer for this constructive comment and for highlighting the relevant work, FedSpaLLM [Bai et al., 2024]. However, we would like to clarify that **this work can be viewed as a specific case within our FedPrLLM framework**. Specifically, this work enables clients to locally prune their models based on private data and send the pruned models to the server for aggregation. The server averages the pruned models using the FedAvg algorithm and prunes the model to satisfy the predefined sparsity rate based on an aggregated mask matrix. This method can be viewed as a specific case within our FedPrLLM framework, i.e., iterative pruning with weight scaling. However, our extensive evaluations (e.g., Table 1 in our paper) reveal that this approach is not optimal.
>
> We hope this explanation addresses the reviewer’s concerns.
>
> > **W2**: Assumption of Public Pre-training: The framework assumes access to a public pre-trained LLM. While standard, the paper does not discuss implications if pre-training data were private—a growing concern in DP-ML.
>
> We thank the reviewer for raising this important point. We would like to clarify that our work operates under the **standard assumption shared by the vast majority of LLM pruning literature** (e.g., Wanda [1], SparseGPT [2], LLM-Pruner [3]): a powerful, pre-trained LLM is already available, and the goal is to compress it efficiently using calibration data.
>
> While the scenario where pre-training data is private is indeed critical, it represents a fundamentally different research problem that focuses on *training* rather than *post-training compression*. Our work specifically targets the practical and immediate need to deploy existing, large-scale open-source models (like LLaMA) in federated environments with limited resources. We hope this explanation addresses the reviewer's concerns.
>
> > **W3**: Calibration Data Efficiency: Each client uses only 2 samples (64 clients × 2 = 128 total). While realistic, the sensitivity to ultra-low calibration data (e.g., 1 sample/client) is not explored.
>
> We thank the reviewer for this insightful question regarding our method's robustness in ultra-low data regimes. To address this, we have conducted new experiments specifically simulating the proposed scenario: 1 sample per client. We ran this challenging experiment on LLaMA-7B and LLaMA-2-7B with 128 clients (each holding only a single calibration sample) at 50% sparsity. For FedPrLLM, we use our recommended configuration of layer comparison, one-shot pruning, and no weight scaling. The results are summarized in the table below.
>
> | Method | LLaMA-7B | LLaMA-2-7B |
> | :--- | :---: | :---: |
> | Centralized | 7.25 /  9.34 / 80.12 |  6.46 / 8.94 / 96.99 |
> | Local-only | 7.58 / 9.73 / 89.22 | 6.77 / 9.30 / 116.60 |
> | FedPrLLM | 7.31 / 9.46 / 82.33 | 6.49 / 9.04 / 103.24 |
>
> *Table A7: Perplexity (WikiText2 / C4 / PTB) of pruned LLMs under 50\% sparsity ratio in the ultra-low data regime (1 sample per client).*
>
> As shown in Table A7, FedPrLLM consistently outperforms the Local-only baseline even in this ultra-low data regime. These results highlight the core strength of FedPrLLM: it effectively aggregates 128 individual masks into a single robust global mask, thereby overcoming the instability that severely impacts the Local-only approach. We thank the reviewer for this suggestion and have incorporated these results into the revised manuscript.

---

> ### Author Response · Authors · 2025-11-22
> **Response to Reviewer 9bhx (2 / 2)**
>
> > **W4**: Communication Cost Nuance: Although one-shot reduces rounds, the paper does not analyze bandwidth vs. latency trade-offs or heterogeneous client capabilities.
>
> We thank the reviewer for this excellent and insightful point. Motivated by this feedback, we have conducted new simulation experiments to provide a more detailed analysis of the trade-offs between bandwidth, latency, and client heterogeneity.
>
> **Bandwidth vs. Latency Trade-off Analysis**: We simulate the total pruning time for one-shot and iterative strategies for LLaMA-7B across four representative network profiles. The results are summarized below:
>
> | Network Profile | Latency | Bandwidth | One-shot Time (h) | Iterative Time (h) | Speedup |
> | :--- | :---: | :---: | :---: | :---: | :---: |
> | Datacenter | 1ms | 10 Gbps | ~0.1 | ~3.1 | ~31x |
> | Cross-Silo (LAN) | 5ms | 1 Gbps | ~1.0 | ~31.3 | ~31x |
> | Cross-Silo (WAN) | 50ms | 100 Mbps | ~9.9 | ~313.3 | ~31x |
> | Edge | 100ms | 10 Mbps | ~99.4 | ~3132.9 | ~31x |
>
> *Table A8: Simulated total communication time under different network conditions.*
>
> As shown in the table above, One-shot pruning achieves a consistent ~31x speedup across all network profiles (from Datacenter to Edge). This demonstrates its robustness to network conditions compared to the iterative baseline, which suffers from the cumulative latency of multiple communication rounds.
>
> **Impact of Client Heterogeneity (Stragglers)**: We simulate a heterogeneous environment where 20% of clients are "stragglers" with 50% of the bandwidth of normal clients (using the Cross-Silo WAN profile). Specifically, we instantiate 64 clients, where 51 *"fast"* clients finish the mask upload in 534.1 seconds, while 13 *"slow"* clients (bandwidth at 50%) take 1,068.2 seconds. In this setting, One-shot pruning incurs a +534 second straggler penalty only once, resulting in a total straggler overhead of 534 seconds (100% of the homogeneous upload time). By contrast, the iterative baseline must absorb the same 534-second penalty at every communication round; with 32 rounds, this compounds to 32 × 534 ≈ 17,090 additional seconds (>4.7 hours) of idle time. This dramatic gap makes One-shot inherently robust to the system heterogeneity typical of cross-device federated learning.
>
> The results demonstrate that **one-shot pruning is clearly superior across all simulated network profiles**, achieving a consistent ~30x speedup compared to the iterative approach. This massive efficiency gain is due to eliminating the need to transmit mask matrices for every layer individually, which is the bottleneck in iterative pruning.
>
> We have included these results in our revision and hope these address the reviewer's concerns.
>
> ----
>
> [1] Mingjie Sun, Zhuang Liu, Anna Bair, and J Zico Kolter. A simple and effective pruning approach for large language models. ICLR, 2024.
>
> [2] Elias Frantar and Dan Alistarh. Sparsegpt: Massive language models can be accurately pruned in one-shot. ICML, 2023.
>
> [3] Xinyin Ma, Gongfan Fang, and Xinchao Wang. Llm-pruner: On the structural pruning of large language models. NeurIPS, 2023.

---

### Official Review · Reviewer_y1jX · 2025-11-05

**Soundness:** 2
**Presentation:** 3
**Contribution:** 2
**Rating:** 2
**Confidence:** 4

**Summary:**

The paper introduces FedPrLLM, a federated pruning framework for large language models (LLMs) aimed at privacy-preserving compression. Clients compute local pruning masks using private calibration data and share them with a server, which aggregates masks to prune the global model. The authors explore three design choices: (1) comparison group (layer, row, column), (2) weight scaling, and (3) pruning strategy (one-shot vs. iterative). Experiments across LLaMA variants and multiple datasets conclude that layer-level comparison, no weight scaling, and one-shot pruning are preferable.

**Strengths:**

- The paper addresses a timely problem: pruning LLMs under privacy constraints in federated settings.
- Provides extensive empirical evaluation across multiple models, sparsity levels, and datasets.
- Clear takeaways (layer comparison, no scaling, one-shot pruning) that practitioners can adopt.
- Framework is simple and easy to implement, making it accessible for real-world experimentation.

**Weaknesses:**

- The approach is a straightforward mask aggregation; similar ideas exist (e.g., FedSpaLLM[1]).
- Weak privacy claim. No secure aggregation or differential privacy. Masks may potentially leak sensitive information
- No comparison to advanced pruning methods (OWL[2], BESA[3], SliceGPT[4]) or structured sparsity approaches. While the authors can argue that some of tghe methods are structured pruning methods, it is important to compare against them.
- Communication costs lack units; no runtime, memory, or speedup metrics; catastrophic perplexities for column comparison unexplained.

**Questions:**

Given the strengths, I have the following questions:

1. The paper positions itself as the first systematic study of federated pruning for LLMs, but prior work (e.g., FedSpaLLM) already explores similar aggregation strategies. Specifically, the \ell_0‑aware aggregation, adaptive mask expansion, layer sampling and related variants from FedSpaLLM that aggregate sparse models or masks to meet global sparsity budgets. How does FedPrLLM fundamentally differ from FedSpaLLM or other federated pruning approaches. The differences that you included in the related work section is not sufficient?

2. The authors noted that previous approaches are not optimal, but there is no comparison against such approaches. The paper limits local pruning to Wanda and SparseGPT but omits competitive post‑training LLM pruning methods that have non‑uniform sparsity (OWL), blockwise reconstruction (BESA), or structured slicing (SliceGPT), each of which has shown superior performance or concrete speedups. Without these baselines, it is unclear whether FedPrLLM’s “best recipe” remains best when local candidates are stronger or structured. Can the authors include these baselines and clarify whether the proposed framework remains competitive.

3. My major concern is in the calculation of the local pruning mask matrix using the private calibration data. When this is sent to the global server, what are the privacy risks involved? Sending the mask without any privacy protection could potentially make the mask vulnerable to attacks.
The paper claims “privacy‑preserving compression” because only binary masks are shared. But masks are data‑dependent and can leak information (presence/absence of features/examples). The FL literature documents strong leakage from shared updates (gradient inversion, membership/property inference). Without secure aggregation or differential privacy analysis, there is no formal protection; moreover, secure aggregation would change communication/computation budgets materially, which the paper does not account for. Can the authors clarify this?

4. I also have a concern around the settings. All experiments shard IID C4 into 128 sequences and hand exactly 2 sequences per client; there is no non‑IID partitioning, no client sampling/dropouts, and no heterogeneity in compute/network, all central to FL. As a result, the conclusions may not transfer to realistic cross‑silo or cross‑device FL deployments. Can the authors clarify this?

5. Communication costs are reported as raw numbers (“6.476B”, “12.952B”) without units (bits? parameters? bytes?). It is relatively unclear what this mean. Also, there is no textual description to clarify this.

6. The catastrophic perplexities for column comparison (e.g., 311,468.53 on LLaMA‑3‑8B at 50% sparsity) indicate numerical instability or a mis‑specified comparison group; the paper does not analyze or mitigate these failures. In addition, Eq. (1) imposes $\|M_\ell\|_0 \ge k$ but the server later enforces exact top‑k (fixed sparsity). The mismatch weakens the formulation.



[1] Bai et al. FedSpaLLM: Federated Pruning of Large Language Models

[2] Yin et al. Outlier Weighed Layerwise Sparsity (OWL): A Missing Secret Sauce for Pruning LLMs to High Sparsity

[3] Xu et al. BESA: Pruning Large Language Models with Blockwise Parameter-Efficient Sparsity Allocation

[4] Ashkboos et al. SliceGPT: Compress large language models by deleting rows and columns

---

> ### Author Response · Authors · 2025-11-22
> **Response to Reviewer y1jX (1 / 5)**
>
> We thank the reviewer for the time and effort in reviewing our paper and providing constructive comments. We are pleased that the reviewer finds the paper to "address a timely problem", provide "extensive empirical evaluation", offer "clear takeaways", and be "accessible for real-world experimentation", all of which inspire us greatly. We have added experiments on advanced pruning methods (i.e., OWL and BESA), privacy analysis, non-IID data distributions, and detailed efficiency metrics, corrected the formulation in Eq. (1), clarified the communication units, and revised the related work section to better position our contribution. Please see our response below regarding the specific comments.
> > **W1 & Q1**: similar ideas exist (e.g., FedSpaLLM[1])
>
> We thank the reviewer for the insightful question and for highlighting the relevant work, FedSpaLLM [1]. We acknowledge that FedSpaLLM presents a valuable contribution to federated LLM pruning. However, our work, FedPrLLM, is **fundamentally different in its objective, scope, and contributions**. While FedSpaLLM proposes a *specific and novel algorithm* for federated LLM pruning, our paper provides the *first systematic and comprehensive empirical study of the fundamental design space* of federated LLM pruning. Our primary goal is not to introduce another single algorithm, but to establish a set of generalizable "best practices" that can guide future research and application in this domain. The key differences are as follows:
> - **Different Research Goals: A Specific Algorithm vs. A Systematic Study**.
>     - FedSpaLLM focuses on proposing a novel algorithmic solution. Its core contributions are specific techniques designed to create a high-performing pruning algorithm. Its methodology, which involves clients sharing locally pruned models to be averaged by the server, can be mapped to a specific configuration within our comprehensive FedPrLLM framework—'iterative pruning with weight scaling'. By evaluating this approach alongside other alternatives, our work complements theirs by providing a valuable benchmark and deeper insight into the effectiveness of the strategy proposed in FedSpaLLM.
>     - FedPrLLM, in contrast, aims to deconstruct the entire federated pruning process into its most fundamental components—specifically, the choice of comparison group, the use of weight scaling, and the pruning strategy (one-shot vs. iterative)—and systematically evaluate their impact. Our contribution lies in the extensive empirical knowledge gained from benchmarking federated pruning for LLMs across a wide range of models, datasets, and hyperparameters. From these efforts, we develop a practical list of key insights for federated pruning of LLMs.
> - **Different Research Questions Investigated**. Our work is structured around three core, unresolved questions that are critical to any federated LLM pruning pipeline, but are not the focus of FedSpaLLM:
>     - Comparison Group: How should parameter importance be compared for aggregation at the server (layer, row, or column)? We find that this choice dramatically impacts performance, with layer comparison as the optimal strategy—a critical finding for practitioners.
>     - Weight Scaling: Should the aggregated mask information be used to scale the retained weights? Our surprising finding is that this intuitive idea offers no benefit and may even degrade performance.
>     - Pruning Strategy: Is a multi-round, iterative pruning process worth the massive communication overhead compared to a simple one-shot approach? We demonstrate that one-shot pruning is equally effective, providing a clear guideline for designing efficient systems.
>
>     These questions are not about *how to build one algorithm*, but about *how the entire research community should build any algorithm*. The insights from our study provide a foundational "recipe" (e.g., "use layer comparison, don't scale weights, and use one-shot pruning") that is simple, effective, and can serve as a strong baseline for future, more complex methods.
> - **Different Contributions to the Field**.
>     - The contribution of FedSpaLLM is a new, competitive algorithm.
>     - The contribution of FedPrLLM is a set of validated, foundational principles and a comprehensive benchmark. We believe our work complements, rather than competes with, works like FedSpaLLM. For instance, our findings on the effectiveness of layer comparison and one-shot pruning could help simplify and improve future iterations of FedSpaLLM or other similar frameworks.
>
> In summary, while both papers address federated LLM pruning, we tackle the problem from different perspectives. FedSpaLLM provides a specific answer, while FedPrLLM provides a map of the terrain and a compass. We have revised our related work section to more clearly express these fundamental differences and better position our unique contribution as the first systematic exploration of this design space. We hope this explanation addresses the reviewer’s concerns.

---

> ### Author Response · Authors · 2025-11-22
> **Response to Reviewer y1jX (2 / 5)**
>
> > **W2 & Q3**: privacy leakage concerns.
>
> We sincerely thank the reviewer for this insightful and critical comment. We acknowledge that our initial claim of "privacy-preserving" was too strong without the formal guarantees we have now provided. Our use of the term stemmed from the foundational principles of Federated Learning (FL), which is inherently designed to enhance privacy by keeping raw data decentralized. However, we agree with the reviewer that even in FL, the *shared updates* (like gradients) can still leak information. A key difference with our FedPrLLM framework is that **we do not share gradients**. Instead, clients only send highly compressed, **binary mask matrices** (just 0s and 1s). This sends significantly less information than standard updates. Since these masks carry so little information, they are naturally much harder to attack. To formally prove this, we perform a detailed privacy analysis on the LLaMA-7B model, covering both theoretical limits and practical attack simulations.
>
> To measure the maximum possible information leakage, we first perform an **Information Entropy Analysis**. This tells us the theoretical limit of how much data a message can hold. Our analysis shows that a binary mask (at 50% sparsity) holds only 1.0 bit of information, while standard Float16 model weights hold 13.75 bits. This means the mask contains only 7.3% of the information found in the weights—a 92.7% reduction. This massive reduction acts as a primary defense, making attack much harder because there is simply very little information available to leak. (*Note: As calculating entropy across all model parameters is computationally prohibitive, this analysis compares data from a single sub-layer (`q_proj`) within the first transformer block.*)
>
> Building on this theory, we test our mask-sharing method with a series of practical experiments. First, we check **Mask Similarity** to see if a mask is uniquely tied to the private data used to create it. We find that masks generated with Wanda using completely different, randomly seeded calibration data are over 95% identical (4.96% Hamming distance). This high similarity proves that the mask matrices are mostly determined by the public pre-trained model's weight, not the private data. This effectively separates the shared information from the private data. Next, our **Differential Privacy (DP) Sensitivity Analysis** shows that changing just one sample in the dataset causes a very small change in the mask matrices (~4.96% Hamming distance). Specifically, we create two datasets that differ by only one sample and measure the difference (i.e., Hamming distance) between their masks. This extremely low sensitivity means our method naturally provides strong privacy protection (equivalent to a formal privacy budget of $\epsilon \approx 0.05$) without needing to add extra noise.
>
> We also simulate targeted attacks to address specific concerns. To test for **Membership Inference Attacks (MIA)** [5], where an attacker tries to guess if a specific data record was used, we simulate a metric-based attack scenario. Since standard MIA relies on confidence scores (which our binary masks don't have), we measure the "signal strength"—the specific influence of a target sample on the final mask. We find that the difference in masks generated with and without a specific target sample is only 3.23% Hamming distance. This variation is smaller than the natural differences caused by using different datasets (~4.96%), making it difficult for an attacker to tell the difference between a real signal and random noise. This implies that any complex attack models would likely fail because the signal is too weak [5,6]. Finally, we simulate a **Gradient Inversion Attacks** [7,8], where an attacker (e.g., an honest-but-curious server) with full knowledge of the model tries to reconstruct the original training data via gradient-based optimization. This attack also fails, recovering less than 2% of the tokens and producing meaningless text. For example:
> -   Original Text: `your Apple AirPods and EarPods. Easy & hassle free installation. Earbuddyz must be removed to charge AirPods...`
> -   Reconstructed Text: `<deuxတTvekirection Readlarzug hecho pertelled hρ threat todos installah="{blearsefw storiesже lookup...`
>
> In summary, by sharing only low-information binary masks, our framework fundamentally reduces the privacy risks compared to sharing gradients. Our comprehensive analysis proves that FedPrLLM offers strong, practical privacy protection. We have updated the paper to include these detailed results and have refined our claims to be more precise. We thank the reviewer for pushing us to strengthen this crucial aspect of our work and hope these added results address the reviewer's concerns.

---

> ### Author Response · Authors · 2025-11-22
> **Response to Reviewer y1jX (3 / 5)**
>
> > **W3 & Q2**: No comparison to advanced pruning methods (OWL[2], BESA[3], SliceGPT[4]) or structured sparsity approaches.
>
> We thank the reviewer for this excellent suggestion. To verify whether our proposed "best recipe" for federated LLM pruning is generalizable and how it interacts with different types of pruning methods, including structured ones, we (1) conduct new experiments with stronger, unstructured pruning methods (OWL and BESA), and (2) provide a clear discussion on the compatibility of our framework with structured pruning.
>
> **1. Generalization to Advanced Unstructured Pruning Methods**
>
> To validate the generality of our findings, we expand our evaluation to include advanced pruning methods like OWL [2] and BESA [3], following the reviewer's excellent suggestion. The results in Tables A1 and A2 confirm that our findings are generalizable. Specifically, our "Best Recipe"—one-shot pruning, layer-wise comparison, and no weight scaling—consistently outperforms other configurations across all methods. This shows that our conclusions extend beyond the initial methods (Wanda/SparseGPT) to represent robust principles for federated LLM pruning.
>
> |Method|Compar. Group|Prune Stra.|Weight Scaling|OWL|BESA|
> |-|-|-|-|-|-|
> |Centralized|-|-|-|7.21/9.31/67.44|7.27/9.34/78.74|
> |Local-only|-|-|-|7.43/9.55/70.11|7.44/9.60/86.19|
> |
> |FedPrLLM|Layer|One-shot|✗|7.24/**9.38**/**67.52**|7.31/**9.43**/**80.28**|
> ||Row|One-shot|✗|**7.23**/9.39/67.56|7.31/**9.43**/80.38|
> ||Column|One-shot|✗|1408.46/914.26/3338.93|1548.53/932.58/4683.50|
> ||Layer|Iterative|✗|**7.23**/9.40/67.77|**7.30**/**9.43**/81.31|
> ||Row|Iterative|✗|**7.23**/9.39/67.62|**7.30**/9.44/81.88|
> ||Column|Iterative|✗|1171.66/905.47/2100.39|1823.51/983.13/4909.44|
> ||Layer|One-shot|✓|7.65/9.87/86.91|7.47/9.63/87.92|
> ||Row|One-shot|✓|7.64/9.87/86.30|7.47/9.64/88.18|
> ||Column|One-shot|✓|1433.32/994.49/3598.38|1693.96/891.77/4662.56|
> ||Layer|Iterative|✓|7.41/9.57/71.36|7.46/9.64/87.44|
> ||Row|Iterative|✓|7.42/9.57/71.24|7.46/9.64/87.64|
> ||Column|Iterative|✓|1455.31/939.69/2790.60|2178.33/1147.38/8064.72|
>
> *Table A1: Perplexity of pruned LLaMA-7B under 50% sparsity ratio using other local pruning methods.*
>
> |Local Compar. Group|Method|Compar. Group|OWL|BESA|
> |-|-|-|-|-|
> |Layer|Centralized|-|7.61/9.88/71.59|7.94/10.28/92.81|
> ||Local-only|-|7.84/10.12/76.15|8.16/10.56/99.17|
> ||FedPrLLM|Layer|**7.80**/**10.12**/**71.76**|**7.98**/**10.34**/**92.26**|
> |||Row|10.54/13.40/124.99|32.54/35.30/291.07|
> |||Column|1115.07/780.56/2480.77|1767.87/966.04/3964.13|
> |
> |Column|Centralized|-|7.89/10.82/72.35|8.23/11.64/100.07|
> ||Local-only|-|7.91/10.86/73.27|8.89/14.19/109.73|
> ||FedPrLLM|Layer|**7.91**/10.84/**73.02**|**8.86**/14.12/**108.12**|
> |||Row|32.24/46.92/645.47|138.87/154.99/1064.28|
> |||Column|**7.91**/**10.83**/**73.02**|**8.86**/**14.10**/108.14|
>
> *Table A2: Perplexity of pruned LLaMA-7B under 50% sparsity ratio when changing the comparison group for the local pruning method. FedPrLLM adopts one-shot pruning and no weight scaling.*
>
> **2. Discussion on Structured Pruning**
>
> We acknowledge the reviewer's important point about structured pruning. However, as stated in our main paper (lines 42-44), **our work currently focuses on unstructured pruning** because it generally achieves higher compression rates and maintains better model performance compared to structured pruning. Our framework is designed to aggregate binary masks for individual weights, which is fundamentally different from structured methods that remove entire substructures like neurons [4,9,10], layers [11], or transformer blocks [12]. These structured methods operate on a different principle that is not directly compatible with our current weight-level voting mechanism.
>
> Nevertheless, we would also like to discuss whether our framework can be extended to structured pruning. Specifically, we believe the core philosophy of FedPrLLM—federated voting on parameter importance—naturally extends to the structural level as "structure-voting". In this paradigm, clients vote on the importance of entire structures (e.g., neurons, layers, or blocks) rather than individual weights. Each client locally identifies a set of structural components to prune (such as specific row or column indices for SliceGPT) and transmits these indices to the server. The server then aggregates these structural "votes", pruning any component marked for removal by a majority of clients (e.g., >50%). This structure-voting approach enables a federated, data-driven method for structured pruning. While a full implementation and evaluation are beyond the scope of this paper, we believe this is a promising direction for future research. Of course, the core question of comparison groups that we explore in FedPrLLM would no longer be applicable in this context. We thank the reviewer for inspiring this valuable discussion and have included it in our future work section.
>
> We hope these new experiments and analyses adequately address the reviewer's concerns.

---

> ### Author Response · Authors · 2025-11-22
> **Response to Reviewer y1jX (4 / 5)**
>
> > **W4-1 & Q5**: Communication costs lack units.
>
> We thank the reviewer for pointing out this ambiguity and apologize for any confusion. The communication costs reported in our paper (e.g., “6.476B”) represent the total number of parameters transmitted between the server and all clients throughout the pruning process, where “B” denotes billions. We have revised the manuscript to explicitly state this in the caption of Table 3 to ensure clarity.
>
> > **W4-2**: no runtime, memory, or speedup metrics.
>
> We thank the reviewer for this constructive comment. To address these concerns, we have conducted comprehensive experiments to quantify runtime, memory usage, and efficiency. The results are presented in the table below.
>
> |Method|Compar. Group|Prune Stra.|Weight Scaling|Pruning Time (s)|Peak Memory (GB)|
> |-|-|-|-|-|-|
> |Centralized|-|-|-|79.8|18.66|
> |Local-only|-|-|-|142.5|25.14|
> |
> |FedPrLLM|Layer|One-shot|✗|143.2|31.27|
> ||Row|One-shot|✗|143.8|31.27|
> ||Column|One-shot|✗|142.9|31.27|
> ||Layer|Iterative|✗|145.6|19.04|
> ||Row|Iterative|✗|144.8|19.04|
> ||Column|Iterative|✗|143.4|19.02|
> ||Layer|One-shot|✓|143.5|31.27|
> ||Row|One-shot|✓|144.1|31.27|
> ||Column|One-shot|✓|143.0|31.27|
> ||Layer|Iterative|✓|144.8|19.69|
> ||Row|Iterative|✓|144.8|19.04|
> ||Column|Iterative|✓|144.8|19.69|
>
> *Table A3: Runtime and Peak Memory usage for all evaluated methods.*
>
> The table shows that the pruning time for One-shot and Iterative methods is similar (~145 seconds), as both involve comparable local computations (Hessian calculation and sorting) on the GPU. The key difference is in communication overhead: One-shot Pruning requires just 1 round of communication (uploading masks once), while Iterative Pruning needs 32 rounds (for LLaMA-7B). Thus, in a real-world federated environment, One-shot pruning will significantly speeds up the total pruning time compared to Iterative pruning due to the drastic reduction in communication rounds. To illustrate this, we simulate the total pruning time under four representative network profiles, as summarized in the table below:
>
> |Network Profile|Latency|Bandwidth|One-shot Time (h)|Iterative Time (h)|Speedup|
> |-|-|-|-|-|-|
> |Datacenter|1ms|10Gbps|~0.1|~3.1|~31x|
> |Cross-Silo (LAN)|5ms|1Gbps|~1.0|~31.3|~31x|
> |Cross-Silo (WAN)|50ms|100Mbps|~9.9|~313.3|~31x|
> |Edge|100ms|10Mbps|~99.4|~3132.9|~31x|
>
> *Table A4: Simulated total communication time under different network conditions.*
>
> As shown in the table above, One-shot pruning achieves a consistent ~31x speedup across all network profiles (from Datacenter to Edge). This demonstrates its robustness to network conditions compared to the iterative baseline, which suffers from the cumulative latency of multiple communication rounds.
>
> Regarding memory usage, One-shot pruning shows a higher peak memory usage (~ 31 GB) compared to Iterative (~19 GB) in our simulation. This is an artifact of our single-machine simulation where the server aggregates masks for all layers simultaneously. In a real distributed deployment, this memory load is distributed across clients, and the server can process masks in a streaming fashion (layer-by-layer aggregation), reducing the peak memory to be comparable to the Iterative approach.
>
> We have included these results in our revision and hope these address the reviewer's concerns.
>
> > **W4-3 & Q6-1**: catastrophic perplexities for column comparison unexplained.
>
> We thank the reviewer for this insightful question, which highlights a critical finding of our work. As analyzed in **Section 4.2.1** of our manuscript, the catastrophic perplexity observed with column comparison arises from a fundamental **mismatch between comparison contexts**. Specifically, our main experiments (Table 1) use a local pruning method with row-wise comparison, meaning clients cast "votes" for pruning based on importance *within each row*. When the server applies column-wise comparison to these row-based votes, it creates a logical flaw, leading to arbitrary pruning decisions that damage the model. Our results in Table 2 are designed specifically to demonstrate this effect, showing that performance is stable only when local and server comparison groups are aligned. Therefore, this finding is a deliberate and central conclusion of our paper: it reveals a critical issue in federated LLM pruning and validates our recommendation to use layer comparison, which consistently avoids this mismatch problem.
>
> > **Q6-2**: Eq. (1) imposes $|M_\ell|_0 \geq k$ but the server later enforces exact top‑k (fixed sparsity). The mismatch weakens the formulation.
>
> We thank the reviewer for their careful reading and for identifying this inconsistency. The constraint in Eq. (1) should be an equality, i.e., $|M_\ell|_0 = k$. The use of '$\geq$' was a typo. We have corrected this in the revised manuscript to ensure our formulation is precise. We appreciate the reviewer's help in improving the rigor of our paper.

---

> ### Author Response · Authors · 2025-11-22
> **Response to Reviewer y1jX (5 / 5)**
>
> > **Q4**: Limited FL settings (IID, no heterogeneity).
>
> We thank the reviewer for raising this important point regarding realistic FL settings. To address these concerns, we have conducted new experiments and analyses covering both **Data Heterogeneity** and **System Heterogeneity**.
>
> **Data Heterogeneity (Non-IID)**: To validate the generalizability of our findings, we further conduct experiments under non-IID conditions. Specifically, we extract 8 samples from the training data of WikiText2, C4, and PTB to form a global calibration dataset (i.e., 24 samples in total). We then use the Dirichlet distribution with a concentration parameter of $\alpha=5$ to split the global calibration dataset into 12 non-IID local calibration datasets, each assigned to one client (i.e., 2 samples per client). We choose Wanda as the local pruning method and use LLaMA-7B to conduct experiments with 50% sparsity pruning. The results are as follows:
>
> |Method|Compar. Group|Prune Stra.|Weight Scaling|LLaMA-7B|
> |-|-|-|-|-|
> |Centralized|-|-|-|7.06/9.27/65.72|
> |Local-only|-|-|-|7.16/9.42/71.54|
> |
> |FedPrLLM|Layer|One-shot|✗|**7.06**/**9.30**/67.54|
> ||Row|One-shot|✗|**7.06**/**9.30**/**67.28**|
> ||Column|One-shot|✗|2923.46/1813.31/6736.30|
> ||Layer|Iterative|✗|**7.06**/9.31/68.09|
> ||Row|Iterative|✗|**7.06**/**9.30**/67.34|
> ||Column|Iterative|✗|3219.96/2294.87/6812.14|
> ||Layer|One-shot|✓|7.17/9.47/72.33|
> ||Row|One-shot|✓|7.17/9.47/72.16|
> ||Column|One-shot|✓|2723.30/1554.46/6364.29|
> ||Layer|Iterative|✓|7.17/9.48/73.40|
> ||Row|Iterative|✓|7.17/9.48/72.92|
> ||Column|Iterative|✓|3182.52/1795.12/5808.61|
>
> *Table A5: Perplexity of pruned LLMs under 50% sparsity ratio using Wanda as the local pruning method under non-IID conditions.*
>
> |Local Compar. Group|Method|Compar. Group|LLaMA-7B|
> |-|-|-|-|
> |Layer|Centralized|-|7.67/10.07/83.20|
> ||Local-only|-|7.76/10.26/85.16|
> ||FedPrLLM|Layer|**7.62**/**10.10**/**81.70**|
> |||Row|43.54/46.29/348.41|
> |||Column|2324.40/1434.18/6026.79|
> |
> |Column|Centralized|-|8.86/14.10/108.37|
> ||Local-only|-|8.86/14.10/108.37|
> ||FedPrLLM|Layer|**8.86**/**14.10**/**108.37**|
> |||Row|138.54/155.15/1060.99|
> |||Column|**8.86**/**14.10**/**108.37**|
>
> *Table A6: Perplexity of pruned LLMs under 50% sparsity ratio when changing the comparison group for the local pruning method (i.e., Wanda) under non-IID conditions. FedPrLLM adopts one-shot pruning and no weight scaling.*
>
> As shown in these results, our proposed "Best Recipe"—using one-shot pruning, layer-wise comparison, and no weight scaling—consistently outperforms other configurations under the non-IID scenario, confirming that our findings are generalizable.
>
> **System Heterogeneity**: For system heterogeneity, we mainly analyze the efficiency of our method. Regarding heterogeneity in network and compute capabilities (e.g., stragglers), the One-shot approach offers a significant structural advantage over iterative methods. To show this, we simulate a system heterogeneity with 20% stragglers (slow clients) to compare the communication time of One-shot and Iterative pruning. Specifically, we instantiate 64 clients, where 51 *"fast"* clients finish the mask upload in 534.1 seconds, while 13 *"slow"* clients (bandwidth at 50%) take 1,068.2 seconds. In this setting, One-shot pruning incurs a +534 second straggler penalty only once, resulting in a total straggler overhead of 534 seconds (100% of the homogeneous upload time). By contrast, the iterative baseline must absorb the same 534-second penalty at every communication round; with 32 rounds, this compounds to 32 × 534 ≈ 17,090 additional seconds (>4.7 hours) of idle time. This dramatic gap makes One-shot inherently robust to the system heterogeneity typical of cross-device federated learning.
>
> We have incorporated these new results and discussions into our revision to strengthen the practical applicability of our work, and hope these address the reviewer's concerns.
>
> ----
>
> [1] Bai et al. FedSpaLLM: Federated Pruning of Large Language Models.
>
> [2] Yin et al. Outlier Weighed Layerwise Sparsity (OWL): A Missing Secret Sauce for Pruning LLMs to High Sparsity.
>
> [3] Xu et al. BESA: Pruning Large Language Models with Blockwise Parameter-Efficient Sparsity Allocation.
>
> [4] Ashkboos et al. SliceGPT: Compress large language models by deleting rows and columns.
>
> [5] Shokri et al. Membership Inference Attacks Against Machine Learning Models.
>
> [6] Dwork et al. Calibrating Noise to Sensitivity in Private Data Analysis.
>
> [7] Zhu et al. Deep Leakage from Gradients.
>
> [8] Fredrikson et al. Model Inversion Attacks that Exploit Confidence Information.
>
> [9] Ma et al. LLM-Pruner: On the Structural Pruning of Large Language Models.
>
> [10] Li et al. LoSparse: Structured Compression of Large Language Models based on Low-Rank and Sparse Approximation.
>
> [11] Xia et al. Sheared LLaMA: Accelerating Language Model Pre-training via Structured Pruning.
>
> [12] Gromov et al. The Unreasonable Ineffectiveness of the Deeper Layers.

---

### Author Response · Authors · 2025-12-01
**Summary Response**

We sincerely thank all reviewers for their insightful comments and constructive feedback. We are encouraged that the reviewers recognize the value of our work, highlighting that it "addresses a timely problem" and "fills a clear research gap" (Reviewer y1jX, qJji), provides "extensive empirical evaluation" that is "rigorous and systematic" (Reviewer y1jX, 9bhx, qJji), offers "clear takeaways" and "counterintuitive insights" (Reviewer y1jX, 9bhx), and makes a "practical and well-executed contribution" with "high practical relevance" (Reviewer 9bhx, XJZZ).

In this rebuttal, we have carefully addressed the concerns raised and significantly strengthened the paper with comprehensive new experiments and analyses:

*   **Distinction from Related Work (Response to Reviewer y1jX, XJZZ)**: We explicitly clarified the distinction between FedPrLLM and prior works (e.g., FedSpaLLM), emphasizing our unique contribution as the first systematic study to map the design space and establish foundational "best practices" for federated LLM pruning.
*   **Generalization to Advanced Methods & Non-IID Settings (Response to Reviewer y1jX, XJZZ, qJji)**: We extended our evaluation to include advanced pruning methods (OWL, BESA) and realistic Non-IID data distributions. Our proposed "Best Recipe" consistently outperforms other configurations in these settings, demonstrating the robustness and generalizability of our findings.
*   **Rigorous Privacy Analysis (Response to Reviewer y1jX, qJji)**: We conducted a formal privacy evaluation, including information entropy analysis, differential privacy sensitivity analysis, and simulated attacks (Membership Inference and Gradient Inversion Attacks). The results quantitatively confirm that our binary mask sharing approach minimizes information leakage and offers strong privacy protection.
*   **Efficiency & System Heterogeneity (Response to Reviewer y1jX, 9bhx, qJji)**: We provided detailed metrics on runtime, memory, and communication overhead. We demonstrated that the One-shot strategy achieves a ~31x speedup and is inherently robust to system heterogeneity (stragglers) compared to iterative methods.
*   **Ultra-Low Data Regime (Response to Reviewer 9bhx)**: We validated the robustness of our method even in extreme data-scarce scenarios (e.g., only 1 calibration sample per client).

We have incorporated these new results into the revised manuscript. We believe these additions make our work a solid and comprehensive guide for the community.

---

### Meta-Review · Area_Chair_JPxh · 2025-12-07

**Summary:**

This paper introduces FedPrLLM, a general federated pruning framework for large language models (LLMs) that allows clients to perform local unstructured pruning and share only binary masks to preserve data privacy. It systematically explores three key design dimensions—comparison group (layer/row/column), weight scaling, and one-shot vs. iterative pruning—across various LLMs, sparsity levels, and datasets. Extensive experiments (6 LLMs, 10 datasets) and derive practical insights (e.g., “layer-wise voting without weight scaling and one-shot pruning works best”) are conducted to investigate the effectiveness.

All reviewers raise critical concerns about the novelty, lack of SOTA comparison, and efficiency analysis. Although the authors partially address several concerns, the main limitation of the novelty and SOTA comparison remains.

The AC has carefully checked the reviews and authors' responses. The AC agrees with the negative points from the reviewers: (1) The work's novelty is limited and borrowed from existing works; (2) The comparisons are not convincing; (3) More evaluations and discussions on real-world settings are required. Therefore, the final recommendation is Reject.

**Reviewer Concerns:**

The authors have partially addressed the concerns:
(1) Distinction from Related Work.
(2) Generalization to Advanced Methods & Non-IID Settings.
(3) Rigorous Privacy Analysis.
(4) Efficiency Analysis.

However, the critical concerns remain:
(1) The main idea is similar to previous works.
(2) More evaluations and discussions on real-world settings are required.

**Reviewer Scores:**

The reviewers may keep their score as the common concerns are not carefully addressed.

---

### Decision · Program_Chairs · 2026-01-26

Reject